

# Modeling the contribution of micronekton diel vertical migrations to carbon export in the mesopelagic zone

Hélène Thibault[1], Frédéric Ménard[1], Jeanne Abitbol-Spangaro[2], Jean-Christophe Poggiale[1], and Séverine Martini[1]

[1]Aix Marseille Univ, Université de Toulon, CNRS, IRD, MIO, Marseille, France
[2]Laboratoire Reproduction et Développement des Plantes, ENS de Lyon, CNRS, Lyon, France

**Correspondence:** Hélène Thibault (helene.thibault@mio.osupytheas.fr)

**Abstract.** Micronekton plays a significant but often overlooked role in carbon transport within the ocean. Using a one-dimensional trait-based model, we simulated the diel vertical migrations of micronekton and their carbon production through respiration, fecal pellets, excretion, and dead bodies. Our model allowed us to explore the biotic and abiotic variables influencing the active transport of carbon in the mesopelagic zone. The functional approach highlighted the importance of size and taxonomy, in particularly considering fish, crustaceans, and cephalopods as key factors controlling the efficiency of carbon transport. Several metabolic parameters accounted for most of the variability in micronekton biomass, organic carbon production, and transport efficiency, mostly linked to respiration rates. Our results suggest that in temperate regions, the export of particles in the mesopelagic zone induced by micronekton is greater in summer, with active carbon transport reaching 18 mgC $m^{-3}$ $y^{-1}$. However, in the context of global warming, the evolution of the impact of micronekton on carbon sequestration remains uncertain. This underscores the imperative for future research to deepen our understanding of micronekton metabolism and vertical dynamics through a functional approach and in relation to their environment.

## 1 Introduction

The ocean regulates the climate by capturing carbon from the atmosphere through biological and physical pumps (Sarmiento, 2006; Boyd et al., 2019). The biological carbon pump (BCP) exports particulate organic carbon (POC) produced primarily by photosynthetic organisms that convert carbon dioxide into organic matter. This generates a downward flux of carbon (Sigman and Boyle, 2000; Boyd and Trull, 2007) composed of marine snow, phytoplankton detritus, zooplankton and nekton fecal pellets, and dead bodies (Turner, 2002). The export of POC is mainly driven by the gravitational sedimentation of a fraction of particles, ultimately sinking below the euphotic zone. Current estimates suggest this export ranges from 4.0 to 9.1 PtC $y^{-1}$ (Siegel et al., 2014). However, recent research revealed the significant contribution of the mesopelagic migratory pump (MMP) to carbon sequestration efficiency (Boyd et al., 2019). Migrant organisms are composed of different size classes including mesozooplankton (0.2–2 mm), macrozooplankton (2-10 mm) and micronekton (10-100 mm). These organisms typically feed near the surface at night and migrate to deeper waters at dawn to avoid visual predators. At these depths, they generate carbon detritus through respiration, egestion of fecal pellets and mortality (Baird et al., 1975; Clarke, 1983; Longhurst et al., 1990).





This ubiquitous movement known as diel vertical migration (DVM) is the Earth's largest animal migration (Hays, 2003).
By transporting carbon actively to the deep ocean, the MMP mitigates particle fragmentation and the remineralization in the surface layers, thereby enhancing the efficiency of the BCP. This contributes to an estimated carbon export ranging from 0.9 to 3.6 PtC y$^{-1}$ (Davison et al., 2013) with a sequestration timescale of approximately 250 years (Boyd et al., 2019).

Previous studies focusing on the MMP mainly looked at zooplankton contribution to the BCP (Steinberg and Landry, 2017). However, recent research revealed that micronekton also plays a significant role in ocean carbon flux (Saba et al., 2021).
These pelagic organisms are described as active swimmers including fish, large crustaceans and cephalopods (Brodeur and Pakhomov, 2019). Despite their lower abundance compared to zooplankton, micronekton can be equally important in driving carbon flux (Pinti et al., 2023). They have greater swimming abilities than mesozooplankton allowing them to go deeper, where they produce larger particles. Additionally, the longer gut transit time of larger migrant organisms allows for the release of fecal pellets after descent into the mesopelagic zone (Pakhomov et al., 1996; Kobari et al., 2008). Whereas the shorter gut evacuation
rate of zooplankton limits their ability to actively transport carbon to depth (Dagg et al., 1989).

Bioenergetic models are used to estimate carbon fluxes induced by micronekton. These fluxes are highly dependent on micronekton biomass, estimated through in situ methods including trawl sampling and acoustic sounders. Acoustic techniques provide a proxy of organism density, allow to distinguish the different scattering layers and their daily dynamics in order to estimate the duration and amplitude of migrations. Trawl net samplings targeting these scattering layers provide data on com-
munity composition, including taxonomy, abundance, and size classes (Koslow et al., 1997). These methods used to estimate micronekton's contribution to the BCP present many uncertainties, mainly due to sampling biases when estimating micronekton biomass using trawling data (e.g., net avoidance, size selectivity) or when using proxies derived from acoustic data. Therefore these uncertainties have a significant impact on the accuracy of carbon flux estimates and on their spatial variability.

Numerous models have been developed to explore the underlying factors driving the dynamics of DVMs and their impacts
on the BCP. Fitness-based models investigate the vertical movements of organisms in the water column by a trade-off between survival, growth and fecundity. These factors vary in response to environmental conditions. For instance, light intensity influences mortality rates through visual predation, while food concentration affects mortality via starvation. Temperature plays a significant role in generation time, with cooler waters typically reducing reproductive and growth rates. Most of the existing modeling approaches primarily focused on copepod migrations (e.g., Fiksen, 1995; Giske et al., 1997; Han and Straškraba,
1998; Eiane and Parisi, 2001; Bandara et al., 2018) or on common fish species such as *Maurolicus muelleri* (e.g., Giske and Aksnes, 1992; Rosland and Giske, 1994, 1997). However, a comprehensive approach considering the entire micronekton community is missing.

Global models are used for estimating the impact of organisms on ocean biogeochemistry based on satellite and in situ ocean observations. However, most of these models focusing on the MMP, typically represent migrant organisms as a single
homogeneous community. These models often assume a uniform distribution of migrant organisms around an isolume of $10^{-3}$ W m$^{-2}$ with migration depths constrained by a tolerance of hypoxia estimated at 15 $\mu$mol L$^{-1}$ (Bianchi et al., 2013; Aumont et al., 2018; Archibald et al., 2019; Nowicki et al., 2022). Consequently, migrant organisms are characterized by a simplified set of parameters, reflecting a lack of comprehensive knowledge regarding the diversity of migratory behaviors and



physiological traits. DVMs are a ubiquitous phenomenon observed across all oceanic regions, yet the depths and velocities of these migrations vary spatially and temporally (Bianchi and Mislan, 2016). This variability in DVM patterns can be attributed to environmental variables affecting organisms' behaviors, community composition, taxonomy, life stage, and size (Frank and Widder, 1997; De Robertis, 2002; Kaartvedt et al., 2009). Despite the widespread occurrence of DVM, information is missing on the relative importance of different taxonomic groups in the active transport of carbon and how environmental variables influence this carbon fluxes across seasons.

Several regional studies investigated the role of micronekton and their migratory dynamics on the BCP. Most of these studies focused on fish species (Davison et al., 2013; Hudson et al., 2014; Belcher et al., 2019; Woodstock et al., 2022; Aksnes et al., 2023), some on large crustaceans (Andersen and Nival, 1991; Pakhomov et al., 2019; Schukat et al., 2013) but few considered the entire micronekton's community (Angel and Pugh, 2000; Hidaka et al., 2001; Ariza et al., 2015; Hernández-León et al., 2019; Kwong et al., 2022; Cotté et al., 2022). These regional studies offer snapshots of specific oceanic processes, yet they frequently overlook temporal variability.

A modeling approach can provide a better understanding of the spatio-temporal dynamics of micronekton, based on in situ observations and models' insights. Micronekton organisms are opportunistic predators that feed on zooplankton including crustaceans, gelatinous, pteropods or small fish (Drazen and Sutton, 2017). These planktivores primarily rely on vision to hunt their prey (Herring, 2001), and take advantage of the contrast created between the illuminated background during the crepuscular periods and the silhouette of their preys (Zaret and Suffern, 1976). Penetration of light in the water column influences daytime depth of residence for micronekton (Frank and Widder, 2002). Furthermore, the visibility of organisms is strongly related to their size (Aksnes and Giske, 1993), however the link between size and depth occupation remains poorly understood for micronekton.

An important source of uncertainty in carbon budget models stems from variations in the physiology of mesopelagic organisms across taxonomic groups, size classes, and environmental conditions. Respiration rates can be estimated by empirical models for different taxonomic groups such as fish, crustaceans, and cephalopods (Ikeda, 2014, 2016; Belcher et al., 2019), accounting for factors like size, temperature, and depth. However, methods are lacking for estimating the production of large, fast-sinking particles, which could contribute significantly to carbon sequestration in the deep ocean.

Some models attempted to address this issue by focusing on plankton size spectra (Kwong et al., 2020; Serra-Pompei et al., 2022) or on migrating behavior (Lehodey et al., 2010). However, a significant gap exists in our understanding of morphological and physiological — that drive the greatest variability in both organic and inorganic carbon production along the water column.

Here, we propose a model to investigate the influence of proximal factors on active carbon transport induced by micronekton. The development of a one-dimensional trait-based model with a limited number of variables offers a valuable tool for testing hypotheses and exploring parameter's sensitivity. While simple models may not fully capture the complexity of interactions between micronekton and their prey, they are an effective way of understanding the influence of some micronekton traits (morphological and physiological) and environmental conditions (e.g., primary production, temperature, light) on carbon export.



This study focuses on characteristic groups of micronekton including fish, cephalopods and large crustaceans of different sizes. The carbon production induced by micronekton was estimated separately for each taxonomic group and by size to investigate their relative impact on carbon transport efficiency in the mesopelagic zone. Sensitivity analyses were conducted to determinate the influence of parameters related to micronekton metabolism. Simulations considering daily and seasonal variations in environmental conditions were conducted to evaluate their effects on the BCP. Environmental variables were derived from a long-term time series site (PAP-SO) in the Northeast Atlantic Ocean. The APERO cruise, which investigates the biological carbon pump including trawl and acoustic samplings, was conducted in this region in June-July 2023. Preliminary biological and environmental data from this cruise provided an opportunity to discuss the potential for calibrating the model to study micronekton contributions to carbon fluxes. Finally, we explored how this model enhances our understanding of the active transport of carbon by micronekton.

## 2  Material and methods

### 2.1  Vertical distribution modeling

The one-dimensional model used in our study represents the diel vertical migration (DVM) and carbon production by micronekton in the water column. The model comprises three state variables represented in Eq.1: the resource (P), the consumer's gut (G), and the consumer's biomass (C), with nine associated parameters displayed in Table 1.

$$
\begin{cases}
\dfrac{\partial P}{\partial t} = \rho Z(1 - \dfrac{P}{K(z)}) - \dfrac{\alpha_v(t,z)P}{1+\beta P}C \\[3mm]
\dfrac{\partial G}{\partial t} = -\dfrac{\partial(wG)}{\partial z} + \dfrac{\alpha_v(t,z)P}{1+\beta P}C - (d+\mu)G \\[3mm]
\dfrac{\partial C}{\partial t} = -\dfrac{\partial(wC)}{\partial z} + edG - (m(t,z)+\mu)C
\end{cases}
\tag{1}
$$

P represents the available prey for micronekton. P is conceptualized as an homogeneous community of mesozooplankton constrained in the surface waters (Fig.1). We assume here that mesozooplankton is not migrating. C corresponds to one of the three taxonomic groups: fish (F), crustacean (A) and cephalopod (S) of average size ranging from 10 to 80 mm, depending on the simulation. These taxonomic groups interact with P when they co-occur at the surface layer. The biomass of the resource varies according to logistic growth dynamics, characterized by a maximal growth rate $\rho$ and a carrying capacity $K$, which is correlated to the phytoplankton concentration.

The transfer of biomass from the resource to the consumer is governed by a Holling type II functional response. Here, instead of being directly assimilated into the consumer's biomass, the ingested food is transferred to the consumer's gut. A proportion



$e$ of the ingested food is then assimilated at a rate $d$, represented by the term $edG$ (see Table.1 for a detailed parameters description).

The numerical method employed to solve this purely advective model is a first-order upwind scheme. This numerical scheme
is chosen for its stability and simplicity, offering first-order accuracy in both space and time. The scheme operates under the Courant–Friedrichs–Lewy (CFL) condition to ensure stability, i.e. $|w|\frac{dt}{dz} \leq 1$, with $dt$ the time step (0.8 h) and $dz$ the depth step (0.2 m).

The swimming speed ($w$ in m h$^{-1}$) is assumed to depend on the swimming abilities of the migrant organisms, their size and the gradient of surface irradiance ($I_0$, modelled in the Supplement),

$$125 \quad w(t) = \frac{w_{max}}{I_0(t)} \frac{dI_0}{dt} \tag{2}$$

where $w_{max}$ corresponds to the maximum swimming speed during the migration phases,

$$w_{max} = a_{swim} \, L \tag{3}$$

where $a_{swim}$ is the swimming coefficient depending on the taxonomic group (see Table.2) and $L$ is the body length (cm).

The visual predation rate $\alpha_v$ varies in response to light penetration in the water column following the Beer-Lambert law,

$$130 \quad I(t,z) = I_0(t) \, e^{-\psi z} \tag{4}$$

where $I(t,z)$ is the irradiance at depth $z$ at a given time $t$ of the day, and $\psi$ is the attenuation coefficient ($\psi = 0.001$). The irradiance ($I$) is then multiplied by a scaling coefficient $c_\alpha$ depending on the simulation (see Table S3 and Fig.S8 in the Supplement).

$$\alpha_v = I(t,z) \, c_\alpha \tag{5}$$

The variation of the saturation parameter ($\beta$=0.01) showed no distinct influence on P and C.

The detritus produced by the metabolic activity of C are composed of fecal pellets ($D_g$), which are the rest of the non-assimilated food, the maintenance products ($D_m$), which represents the production of carbon expended by the consumer for maintaining his physiological functions (more details in the next section), and the dead bodies ($D_\mu$),

$$\begin{cases} \dfrac{\partial D_g}{\partial t} = (1-e)dG - r(T)D_g \\[2em] \dfrac{\partial D_m}{\partial t} = m(t,z)C + edG \, c_{sda} \\[2em] \dfrac{\partial D_\mu}{\partial t} = \mu(C+G) \end{cases} \tag{6}$$




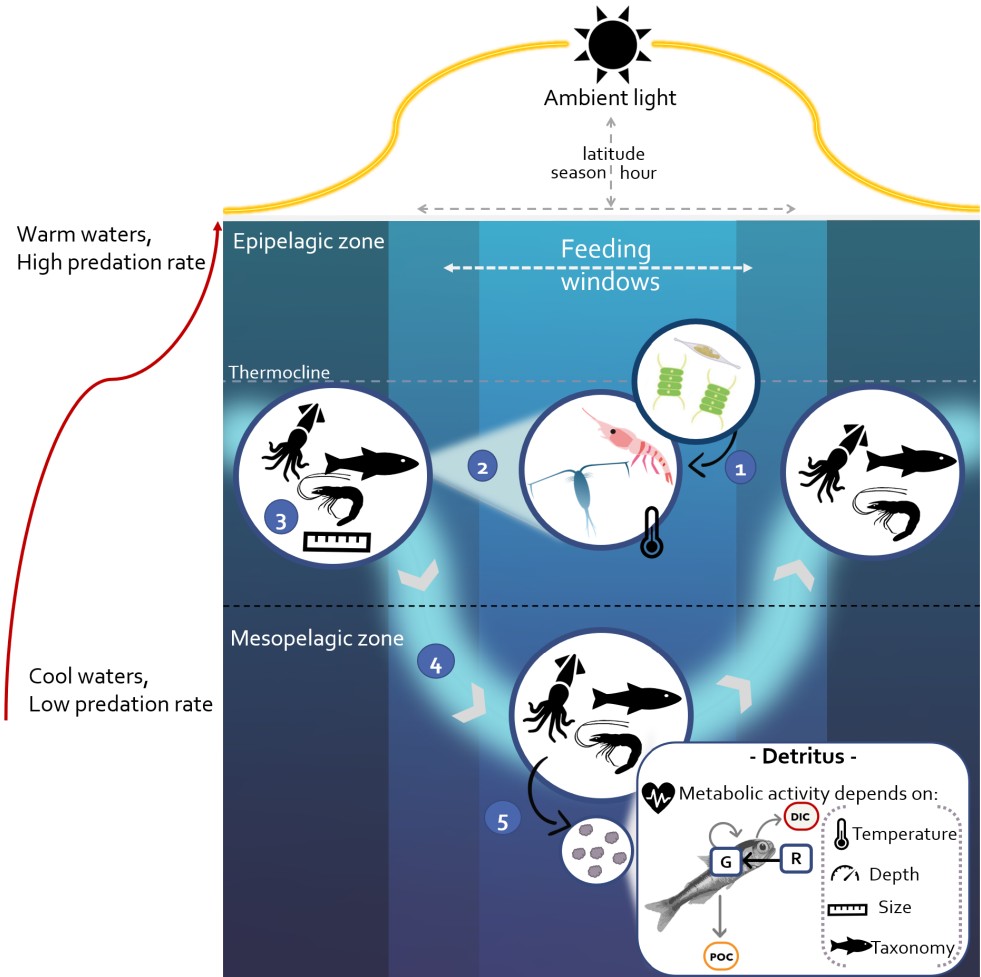

**Figure 1.** Conceptual diagram of the model. 1) Zooplankton growth depends on phytoplankton concentration and is temperature dependent. 2) Micronekton uses visual predation to hunt zooplankton. 3) Simulations are set for one taxonomic group (fish, cephalopod or crustacean) for a given size. 4) DVM is triggered by light rate of change. Migration speed depends on light variation, size and taxonomy. 5) The resource (P) ingested is transferred to the gut (G). One part is assimilated for growth and the other part is released as fecal pellets. Micronekton produces detritus along the water column, composed of dissolved inorganic carbon (DIC) via the respiration ($D_m$), the particulate organic carbon (POC) including fecal pellets ($D_g$) and dead bodies ($D_\mu$).

## 2.2 Bioenergetic model

### 2.2.1 Metabolic costs

Energy costs associated with micronekton metabolism can be estimated based on the rate of respiration, reflecting the catabolic processes of individuals. Micronekton organisms exhibit varying metabolic rates depending on their activity levels throughout



**Table 1.** Micronekton model parameters from the literature for the different taxonomic groups: Resource (P), Fish (F), Crustacean (A) and Cephalopod (S), and the range associated when available in the references. Type corresponds to the taxon associated with the parameter in the study cited (Source) and group corresponds to one of the three taxonomic groups used in the simulations.

| Parameter | Symbol | Value [range] | Unit | Type | Group | Source |
|---|---|---|---|---|---|---|
| Max. growth rate | $\rho$ | 0.0062 | $h^{-1}$ | Zooplankton | P | Jager and Ravagnan (2016) |
| Fraction of PP to zooplankton | $\varepsilon_{PP,Z}$ | 0.32 [0.25–0.47] | - | Zooplankton | P | Anderson et al. (2019) |
| Trophic efficiency (%) | $\gamma$ | $10.13 \pm 5.81$ | - | Fish | F,A,S | Pauly and Christensen (1995) |
| Gut evacuation rate | $d$ | 0.67 | $h^{-1}$ | Decapoda | A | Beseres et al. (2006) |
| | $d$ | 0.5 | $h^{-1}$ | Cephalopod | S | Lipiński (1987) |
| | $d$ | 0.2 | $h^{-1}$ | Myctophidae | F | Pakhomov et al. (1996) |
| Assimilation efficiency | $e$ | 0.8 | - | *Maurolicus muelleri* | F | Ikeda (1996) |
| | $e$ | 0.9 | - | Cephalopod | S | Boyle and Rodhouse (2008) |
| | $e$ | 0.8 | - | Crustacean | A | Lasker (1966) |
| Taxonomic ratio | $\delta_f$ | 0.7 | - | Fish | F | Ariza et al. (2015) |
| | $\delta_s$ | 0.2 | - | Cephalopod | S | Ariza et al. (2015) |
| | $\delta_c$ | 0.1 | - | Crustacean | A | Ariza et al. (2015) |
| Asymptotic length | $L_\infty$ | 10.24 [3.3-26.5] | cm | Mesopelagic fish | F | Pauly et al. (2021) |
| Growth coefficient | $k$ | 0.53 [0.17-5.62] | $y^{-1}$ | Mesopelagic fish | F | Pauly et al. (2021) |
| Mortality coefficient | $\lambda$ | 5.26 | - | Zooplankton | A | Zhang and Dam (1997) |
| SDA coefficient | $c_{sda}$ | 0.14 [0.12-0.16] | - | Fish | F,A,S | Brett et al. (1979) |
| Activity factor of swim | $A_m$ | 4 [1-4] | - | Fish | F | Brett et al. (1979) |
| | $A_m$ | 3 [1-3] | - | Crustacean | A,S | Torres and Childress (1983) |
| Respiratory quotient | $Q_R$ | 0.9 [0.7-1] | - | Fish | F | Brett et al. (1979); Hudson et al. (2014) |
| | $Q_R$ | 0.97 [0.6-1.61] | - | Crustacean | A | Mayzaud et al. (2005) |
| | $Q_R$ | 0.95 | - | Cephalopod | S | Birk (2018) |

the day. First, the basal activity is composed of the standard metabolic rate (SMR), when the organisms are at rest and the routine metabolic rate (RMR), corresponding to a minimal activity, when the micronekton swims locally. Secondly, the active metabolic rate (AMR) corresponds to a high catabolic activity, when the organisms swim actively, typically during the migration phases. Finally, the specific dynamic action (SDA) represents the energy expenditure associated with feeding-related activities, including the digestion and assimilation of ingested food (see Eq.6).

The maintenance rate (m in $h^{-1}$) in Eq.6, is expressed as the sum of the RMR ($R_C$) and the AMR ($R_S$),

$$m = R_C(t,z) + R_S(t,z) \tag{7}$$

The SDA is proportional to the amount of food assimilated, expressed in Eq.6 by the term $edG\ c_{sda}$, which corresponds to 14% of assimilated amount (Brett et al., 1979).



**Table 2.** Taxonomic-dependent parameters of the model for the three different consumers including fish, crustaceans and cephalopods. More details about their range and the citation associated are displayed in Table.1.

| Group | $a_0$ | $a_1$ | $a_2$ | $a_3$ | $Q_R$ | $d$ | $e$ | $a_{swim}$ | $\delta$ |
|---|---|---|---|---|---|---|---|---|---|
| Fish | 30.767 | 0.870 | -8.515 | -0.088 | 0.9 | 0.2 | 0.8 | 1.8 | 0.7 |
| Crustacean | 23.097 | 0.813 | -6.248 | -0.136 | 0.97 | 0.4 | 0.8 | 1.1 | 0.1 |
| Cephalopod | 24.461 | 0.868 | -6.424 | -0.261 | 0.95 | 0.5 | 0.9 | 1.25 | 0.2 |

For crustaceans, metabolic costs represent losses due to respiration and excretion while the amount of DOC produced by fish can be negligible (Davison et al., 2013; McMonagle et al., 2023). Cephalopods are not known to excrete DOC either, and the production of mucus has never been quantified (Boyle and Rodhouse, 2008). Excretion of DOC for crustaceans represents 31% of $CO_2$ respired (Steinberg et al., 2000). For simplification, the part of DOC produced by crustaceans was added to the respiration in the results as the part of dissolved carbon production ($D_m$). RMRs ($\mu L\ O_2\ ind^{-1}\ h^{-1}$) of pelagic marine fish, crustaceans and cephalopods are expressed as a function of body mass, temperature and depth (Ikeda, 2016, 2014). We used a multiple regression model, with coefficients varying by taxonomic groups (see Table.2).

$$lnR = a0 + a1\ lnM_C + a2(1000/T) + a3\ lnD \tag{8}$$

where $M_C$ is the body mass (mgC), $T$ the habitat temperature ($K$) and $D$ the depth (m). The following equation is applied to convert in a carbon production rate ($R_C$ in h$^{-1}$),

$$R_C = \frac{R}{M_C}\ Q_R\ \frac{12}{22.4} \tag{9}$$

where 12/22.4 is the mass of carbon (12 g) in 1 mol (22.4 L) of $CO_2$ and $Q_R$ is the dimensionless respiratory taxonomic-dependent quotient (Table.2).

During migrations, swimming activity generates a catabolic cost. According to Brett et al. (1979), the AMR is equal to four times the basal metabolic rate for fish. Here, we derived it from the RMR, equivalent to twice the SMR, when migrant organisms reach their maximum swimming speed ($w_{max}$). This AMR ($R_S$) is calculated from a linear relationship between swimming speed and basal metabolic rate proposed by Torres and Childress (1983):

$$R_S = \frac{w_{swim}}{w_{max}} 2R_C \tag{10}$$

### 2.2.2 Mortality rate

The mortality rate ($\mu$ in d$^{-1}$) was calculated differently according to the taxonomic group. For marine fishes, the mortality rate is defined as an allometric relationship (Lorenzen et al., 2022).

$$ln\mu = \frac{0.28 - 1.30\ log(L/L_\infty) + 1.08\ log(k)}{365} \tag{11}$$





with $L$ the body length (cm), $L_\infty$ and $k$ the parameters from Von Bertalanffy estimated from an average of Myctophidae's values available in Fishbase (Pauly et al., 2021).

For crustaceans, the mortality rate is defined from Zhang and Dam (1997), adapted from Peterson and Wroblewski (1984),

$$\mu = \lambda \, 10^{-3} W_D^{-0.25} \tag{12}$$

where $W_D$ is the dry weight (g) and $\lambda$ the mortality coefficient (Table.1).

All cephalopods except Nautilidae are known to have a particular life cycle with a single episode of reproduction leading to the death of the organism. The lifespan is one or two years which corresponds to an average mortality rate around $1.8 \times 10^{-3}$ $\mathrm{d}^{-1}$.

## 2.3  Modeling environment variations

The model environment is characterized by several key variables, including temperature, phytoplankton concentration, and
light availability, all of which vary with depth, on a daily basis, and over the course of the year. These environmental factors play crucial roles in shaping the distribution, behavior, and physiology of micronekton organisms, as developed in the following sections.

### 2.3.1  Temperature-dependent process

The fecal pellet remineralization rate ($r$ in $\mathrm{d}^{-1}$) is assumed to be temperature-dependent and calculated from a linear relation-
ship used in Brun et al. (2019),

$$r = 0.005T + 0.011 \tag{13}$$

where $T$ is the temperature (°C).

The effects of temperature on physical and physiological processes are implemented as factors affecting the growth rate of zooplankton. The factor $Q_{10}$ is used to model the effects of temperature on each corresponding parameter based on the Van't
Hoff rule, which states that a change of the temperature $T$ by 10° will multiply the rate $\rho$ at the reference temperature $T_{ref}$ by a factor $Q_{10}$,

$$\rho = \rho_{ref} \, Q_{10}^{(T-T_{ref})/10} \tag{14}$$

where $T$ is the temperature and $\rho_{ref}$ the rate at the reference temperature. A $Q_{10} = 3$ is used for the physiological processes of zooplankton (Hirst et al., 2003) with $\rho_{ref} = 0.008 \ \mathrm{h}^{-1}$ and $T_{ref} = 15°C$ as reference values. The effect of temperature on
the gut evacuation rate was not taken into account, as it induces a negligible variation for the temperature interval considered in this study, based on the relationship established in Pakhomov et al. (1996).

### 2.3.2  Modeling irradiance

Surface irradiance ($I_0$) is modelled to define the rate of change of light that triggers DVMs (Fig.2).





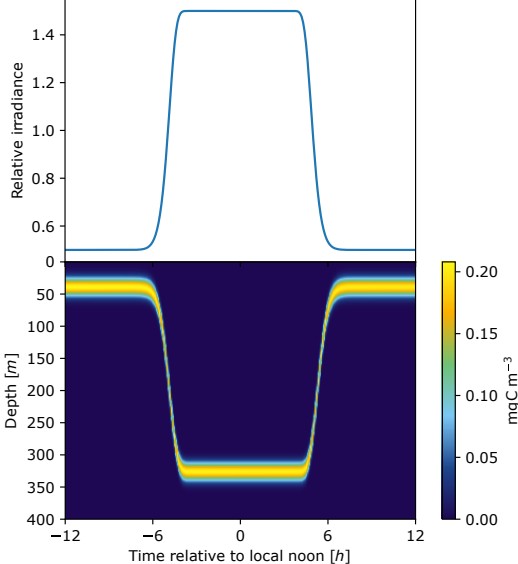

**Figure 2.** Surface irradiance on top is used as a trigger of DVM that define the relative migration speed. The time of descent corresponds to a positive speed and ascent a negative speed during the crepuscular periods. This results in a pattern of diel vertical migration over 24h as an example here for a community of fish measuring 40 mm.

The light is then modelled in depth to constrain the visual capture rate (see Eq.5), as a function of chlorophyll concentration.
All the equations used to model irradiance at the surface and at depth are described in section 2 of the Supplement.

### 2.3.3 Environmental data at PAP-SO (48° 50'N, 16° 30'W)

Environmental conditions are set from the Porcupine Abyssal Plain Sustained Observatory (PAP-SO at 48° 50'N 16° 30'W, 4,850 m water depth). It is part of the North Atlantic Drift, characterised by a large spring and secondary late summer/autumn blooms generating a high export of carbon, with light and/or nutrient limitation during the rest of the year (Sutton et al., 2017).
All data were interpolated to fit the simulation grid for one year with a time step of 0.8 h from the surface to 1,000 m depth with 0.2 m interval.

Temperature data, which contains the 3D monthly mean potential temperature with depth resolution were collected from Global Ocean Physics Analysis and Forecast, E.U. Copernicus Marine Service Information (CMEMS), Marine Data Store (MDS). https://doi.org/10.48670/moi-00016
Mole concentration of phytoplankton expressed as carbon in sea water (PHYC) was collected from the Atlantic-Iberian Biscay Irish- Ocean Biogeochemical Analysis and Forecast, E.U. Copernicus Marine Service Information (CMEMS), Marine Data Store (MDS). https://doi.org/10.48670/moi-00028





The IBI36 system provides a depth simulated pythoplankton concentration expressed as carbon per unit volume in sea water compared to four algorithms (VGPM, Eppley-VGPM, CbPM and Cafe), built on the satellite data. Mole concentrations (mmolC m$^{-3}$) were converted in carbon mass (mgC m$^{-3}$).

## 2.4 Sensitivity analysis

A sensitivity analysis of model's outputs was conducted to assess the impact of metabolic parameters for each taxonomic group (fish, crustacean, cephalopod) on carbon production and transport. Indeed, metabolic parameters were compiled from the literature and used in the empirical relationships (Eq.8,11,12), allowing to test the influence of the variability of metabolic costs on carbon transport. Simulations were carried out separately for each taxonomic group, assuming a standardized size of individuals of 30 mm. The sensitivity of the model parameters focused on three key model outputs including 1) micronekton's biomass, potentially linked to the induced carbon production by micronekton, 2) the POC production comprising fecal pellets and dead bodies that sediment and sequester carbon at depth in the water column, and 3) the efficiency of POC transport below the euphotic zone ($pe-ratio_{200}$). This metric is calculated as the total production of POC under 200 m depth divided by the integrated phytoplankton concentration in the surface layers. This allows to investigate the link between the estimation of the biomass of micronekton and their efficiency to transport carbon, which originates from the primary production at the surface.

The model parameters are sourced from the literature and their ranges (Table.1), with standard deviations associated for the respiration coefficients (Table.S2 in the Supplement). In cases where ranges were not available, a variation of 25% is applied, according to the range observed for other parameters.

Bioenergetics equations and associated parameters are selected based on size and taxonomic groups, including fish, crustaceans, and cephalopods. Parameters are chosen from the most common micronekton taxa whenever possible. For instance, Myctophidae is the most widespread family of micronekton fish performing diel vertical migration (DVM). Micronekton crustaceans primarily consist of shrimps and large euphausiids, while migrant cephalopods belonging to micronekton are predominantly composed of squids and octopus. These taxonomic considerations are taken into account when selecting appropriate parameters to represent the bioenergetic processes in the model.

Sobol's indices are used to quantify the relative contributions of the metabolic parameters to the model's outputs (Sobol, 2001; Saltelli, 2002; Saltelli et al., 2010). The Sobol's sequence allows to generate set of parameters with uniform partition within their range, allowing low-discrepancy Monte Carlo simulations. Total Sobol's indices ($S_{Ti}$) were computed to represent the total contribution of each input variable to the overall variance in the model's output, accounting for both direct and interaction effects (Eq.15). These indices range from zero to one, with one indicating the total variance of the model output.

$$S_{Ti} = 1 - \frac{Var(E(Y|X_\theta))}{Var(Y)} \tag{15}$$

where $Var(E(Y|X_\theta))$ is the variance of the expected value of the model output $Y$ when input variable ($X_i$) is excluded and all other variables are included ($X_\theta$), and $Var(Y)$ is the total variance of $Y$.



A total of 12 parameters were tested, resulting in 1536 simulations for each of the three taxonomic groups. To ensure the
robustness of the results, the sensitivity analysis was run twice and produced consistent outcomes. The Global Sensitivity
Analysis features provided by SALib (Herman and Usher, 2017; Iwanaga et al., 2022) facilitated the computation of Sobol's
indices.

## 2.5 Size and taxonomic-dependent simulations

The model is executed individually for each taxonomic groups (F, A, S) across various sizes of individuals ranging from 10
to 80 mm, with the associated parameters listed in Table.2). Specifically, a range between 20 mm and 80 mm is employed for
fish and cephalopods, while a range between 10 mm and 50 mm is used for crustacean. This approach ensured that the model
encompassed a diverse range of sizes representative of migrant micronekton organisms including fish and crustaceans, based on
night epipelagic samples (e.g., Kwong et al., 2018), allowing for a comprehensive assessment of the metabolic impacts across
different taxonomic groups and size spectra. Size spectrum is challenging to assess for cephalopods (different measurement
method, sample damage). Thus, the same size range as fish is applied, allowing comparison.

The capture rate coefficient ($c_\alpha$) is calibrated to ensure a transfer efficiency ($\gamma$) of 10% between the biomass of consumers
and their resource (Table.1). This transfer efficiency was further adjusted using a taxonomic ratio ($\delta$), which allows for the
estimation of the relative biomass distribution among fish, crustaceans, and cephalopods within the micronekton community.

$c_\alpha$ is estimated for each simulation, allowing the calibration for P and C concentrations. This calibration varies according to
size, taxonomic group, and seasonal simulation. More details about this method are provided in the Supplement (see Table S3
and Fig.S8).

## 3 Results

### 3.1 Micronekton's traits and carbon production

#### 3.1.1 Influence of size and taxonomy on carbon production

Independent simulations involving one taxonomic group of a given size were conducted to study their impact on carbon produc-
tion by micronekton, without seasonal variation of the environment. All migrant organisms of different size (fish, crustaceans
and cephalopods) exhibited surface feeding behavior during the night, where they consumed food resources (Fig.1). A portion
of the ingested food was retained in their gut and subsequently released at their daytime depth of residence as fecal pellets
(Fig.3a-c). This resulted in a peak in metabolic activity at depth during daytime, due to the specific dynamic action (Fig.S1 in
the Supplement) and a peak at the surface during the nighttime (Fig.3a-l).





**Figure 3.** Carbon annual production along depth (mgC m$^{-3}$ y$^{-1}$) represented by taxonomic groups: Fish (a, d, g), Crustacean (b, e, f), and Cephalopod (c, f, i), and by size (10-80 mm) indicated by the colorbar. Carbon detritus include fecal pellet production $D_g$ (a, b, c), released of carbon dioxide and dissolved organic carbon $D_m$ (d, e, f), and dead bodies $D_\mu$ (g, h, i). The bottom panels represent the integrated carbon production along depth and the efficiency of POC transfer below 200 m depth ($pe - ratio_{200}$) as a function of size, respectively for fish (j), crustaceans (k) and cephalopods (l). Each point or each curve represents a single simulation for one taxonomic group of a given size.



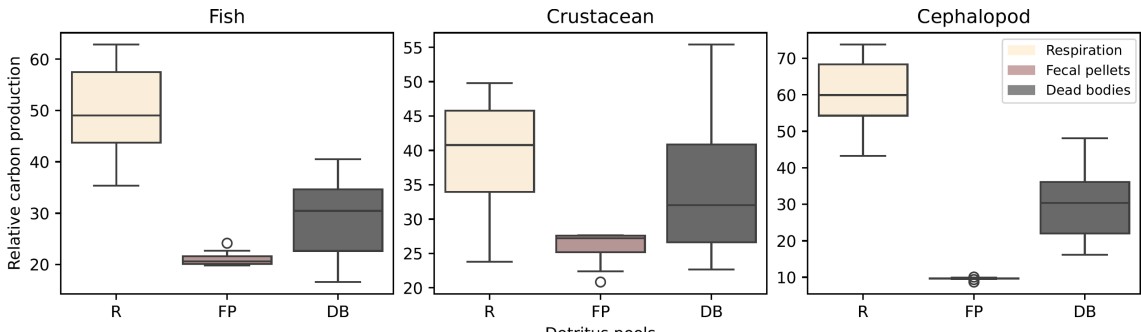

**Figure 4.** Relative contribution of carbon detritus produced by the three taxonomic groups of micronekton (fish, crustacean, cephalopod) including the respiration $D_m$ (R), the fecal pellets $D_g$ (FP) and the dead bodies $D_\mu$ (DB). The variability is due to the size of the organisms, with the same size ranges as in Fig.3.

Each migrant organism spent as much time at the surface as it did at depth, and took about an hour to descend from the surface to the maximum migration depth (see an example for a fish of 35 mm in the Supplement, Fig.S3).

Globally, a greater proportion of carbon production was released at the surface through fecal pellets ($D_g$), metabolic products ($D_m$) and dead bodies ($D_\mu$), where the organisms reside during the night (Fig.3). Regarding migration depths during the day,
fish ranged between 200 and 700 m depth (Fig.3a,d,g), from the smallest to the largest individuals. Crustaceans and cephalopods resided to shallower depths during the day, respectively between 100-300 m (Fig.3b,e,h) and 150-500 m (Fig.3c,f,i).

In Fig.3, each curve corresponds to equivalent concentrations within the same taxonomic group, but is composed of individuals with different size. Both size and taxonomy had an impact on the ability of organisms to export carbon. For each taxonomic groups, smaller individuals (10 mm for crustaceans or 20 mm for fish and cephalopods) migrated to shallower depth, above
200 m depth (Fig.3). Size also influenced each of the three carbon production pools ($D_g$, $D_m$, $D_\mu$) induced by fish, crustaceans and cephalopods. Smaller individuals produced more carbon through $D_g$, $D_m$ and $D_\mu$ for each taxonomic group (Fig.3). Size affected the production of $D_\mu$ to a greater extent, with smaller individuals having higher mortality rates. To a lesser extent, size also influenced the production of $D_g$, which was higher for smaller individuals and also, less significantly, for $D_m$.

Based on the taxonomic-ratio of the model (Table.1), fish was the most abundant group of micronekton (70%), followed
by cephalopods (20%) and crustaceans (10%). Fish produced more carbon detritus than the other groups and at deeper depth layers (Fig.3). This resulted in a global higher $pe-ratio_{200}$, i.e. the ability of organisms to export carbon below the epipelagic zone (Fig.3j). The $pe-ratio_{200}$ of crustaceans smaller than 30 mm was null, as they did not reach 200 m depth, and the same was true for cephalopods of 20 mm. The $pe-ratio_{200}$ reached its maximum values for intermediate sizes, i.e. 31 mm for fish, 42 mm for crustaceans and 41 mm for cephalopods (Fig.3j,k,l). Larger crustaceans between 35 mm and 50 mm generated a
higher $pe-ratio_{200}$ than fish larger than 50 mm. Crustaceans were less abundant than cephalopods but they produced more $D_g$ and $D_\mu$. However, cephalopods produced higher $D_m$ than crustaceans with an overall higher $pe-ratio_{200}$. Cephalopods had much smaller natural mortality rate, particularly for smaller individuals than the other groups.





The relative contribution of carbon detritus varied among each of the three taxonomic groups (Fig.4, Table S1 in the Supplement). On average, metabolic products accounted for 50% of the carbon metabolized by fish, 40% for crustaceans (which also included the excretion of DOC), and 60% for cephalopods. The average relative production of dead bodies was similar among the groups, around 30%. However, this proportion varied greatly with size for crustaceans, from 55% for smaller individuals to 23% for larger ones. Cephalopods produced low amounts of fecal pellets, representing 10% of their carbon production, while it was more than 20% for fish and 27% on average for crustaceans.

### 3.1.2 Sensitivity analysis on metabolic parameters

A sensitivity analysis was performed on the metabolic parameters of micronekton (Fig.5). Total Sobol's indices were calculated to assess the influence of these parameters on micronekton's biomass, total POC production, and $pe - ratio$ below 200 m.

Overall, the parameters showed similar sensitivity across different model outputs, with slight variations observed within taxonomic groups. In particular, the respiration coefficients $a_0$ and $a_2$ exerted the greatest influence on the outputs of all taxonomic groups (Fig.5), especially for cephalopods and fish. Other sensitive parameters associated to the respiration rate were $a_{swim}$, and $Q_R$ only for crustacean.

The parameters $a_{swim}$ associated to the swimming speed (Eq.3) generated a high variability in the model outputs, particularly for the $pe - ratio_{200}$ and for crustacean. The gut evacuation rate $d$ had also a slight influence on the $pe - ratio_{200}$ for crustaceans, which was not the case regarding the biomass and the POC, as for the other taxonomic groups. The assimilation efficiency $e$ was mostly sensitive for crustaceans and cephalopods regarding the POC production.

In the case of fish, Sobol's indices were very similar between biomass, POC and $pe - ratio_{200}$. Mortality parameters ($L_{inf}$ and $k$) were also slightly sensitive.

### 3.2 Environment variability and carbon production

The environment was represented by three variables including phytoplankton concentration, temperature and irradiance (Fig.6), in order to study their impact on the carbon production by micronekton.

Most of the primary production was concentrated in the first 50 m. The maximum of phytoplankton concentrations occurred in April-May and reached 42 mgC m$^{-3}$ at the surface. Sea surface temperature ranged from 12°C in winter to 19.5 °C in summer. The greatest temperature variability occurred in the first 100 m. At the winter solstice in late December, days lasted only 6 hours, and irradiance was the lowest of the year. In contrast, during the summer solstice in June, daylight lasted 12 hours and the relative surface irradiance reached its maximum.

One peak of carbon production induced by micronekton occurred during the year in June (Fig.7a). This peak reached over 40 mgC m$^{-2}$ d$^{-1}$. Carbon concentrations then fell, reaching their lowest values in the winter of 7 mgC m$^{-2}$ d$^{-1}$. The proportion of POC transported from the surface to the mesopelagic zone by micronekton, calculated as the $pe - ratio_{200}$, varied greatly during the year. From 0.5 $10^{-3}$ in January to 6.5 $10^{-3}$ in September (Fig.7b).

The proportion of fecal pellets in carbon production induced by micronekton oscillated during the year. This proportion reached its maximum in March of 20% and its lowest values in July-August of less than 10%, when temperatures were highest





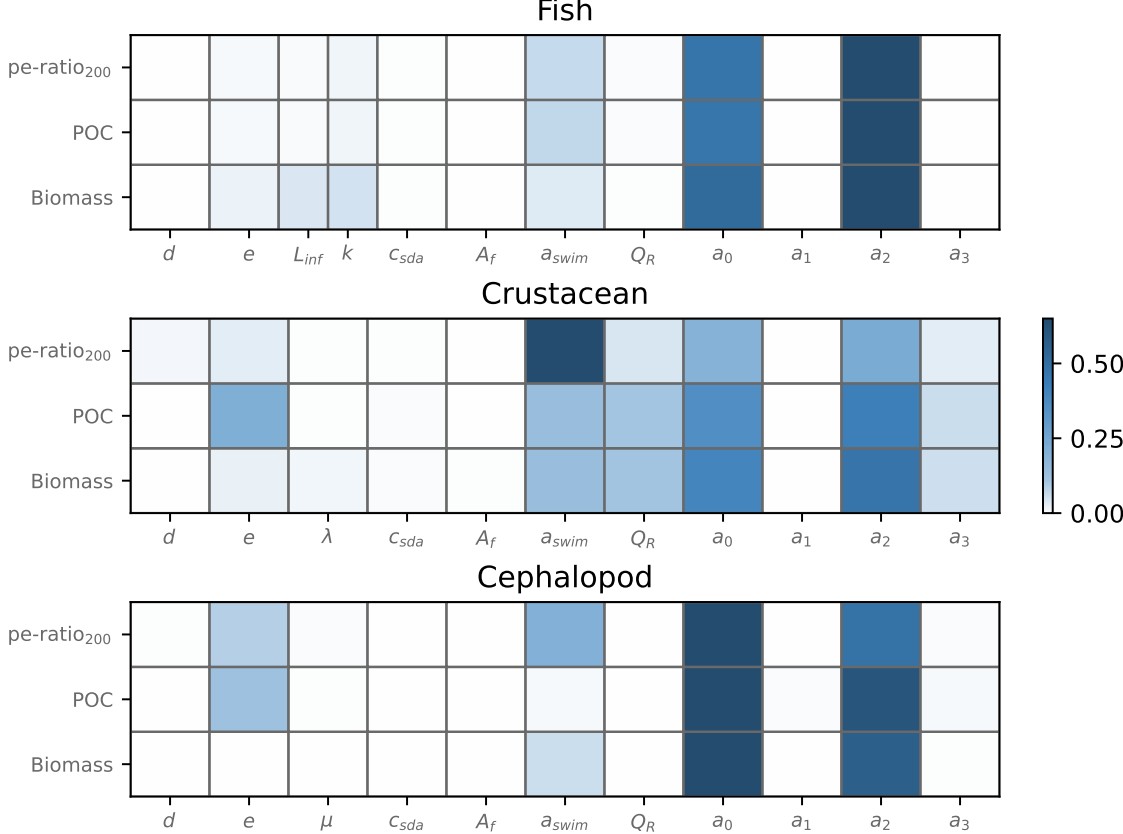

**Figure 5.** Total Sobol's indices generated separately for the three taxonomic groups on different model outputs: total biomass of the consumers (Biomass), total production of fecal pellets and dead bodies (POC) and pe-ratio representing the efficiency of POC transport under 200 m depth ($pe - ratio_{200}$). Organism size was set at 30 mm for each simulations.

(Fig.7b). The proportion of respiration varied slightly from 55% the rest of the year to almost 60% in September. The proportion of dead bodies varied inversely with the proportion of fecal pellets in the carbon production.

Seasonal variations in surface irradiance had a major impact on the migration depths of organisms (Fig.6c; Fig.8). Daytime residence depths varied between 220 m in winter to 400 m in summer for a homogeneous community of fish of measuring 335    35 mm. During summer, micronekton produced greater quantities of carbon at greater depths than the rest of the year. Fecal pellet production was lower in autumn and winter, when micronekton concentrations are lower (Fig.S6 in the Supplement), temperatures colder and days shorter (Fig.6).





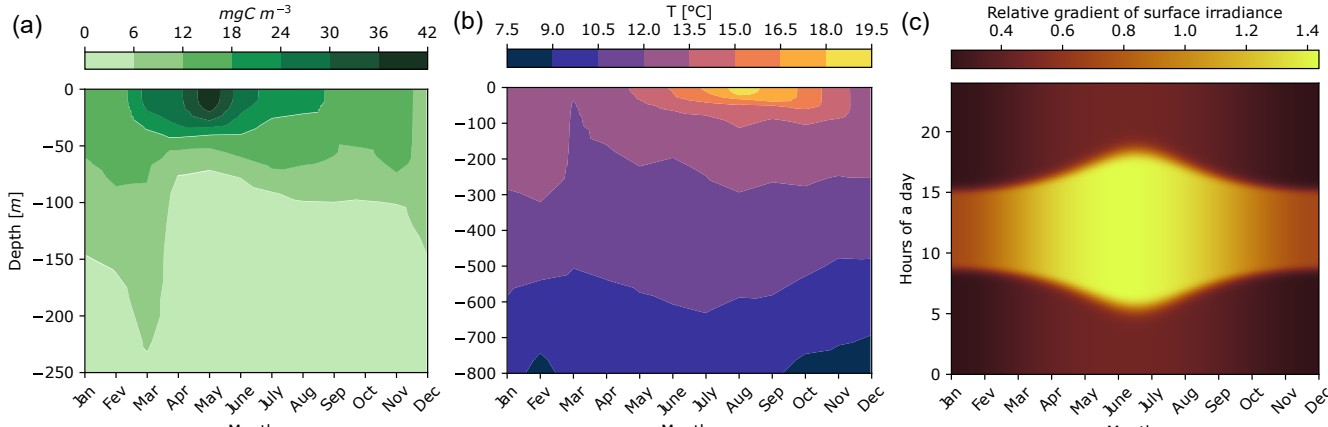

**Figure 6.** Seasonal variation of the environmental conditions at PAP-SO ($48°$ 50'N, $016°$ 30'W) including monthly mean of phytoplankton concentrations (a), monthly mean of temperatures (b) and the modeled surface irradiance (c) calculated in the Supplement.

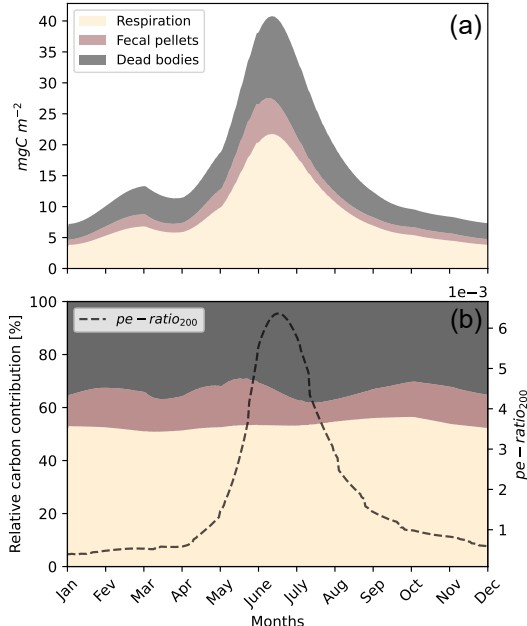

**Figure 7.** Seasonal variation of carbon detritus including the metabolic products ($D_m$) and POC production via fecal pellets ($D_g$) and dead bodies ($D_\mu$), integrated along depth (a), and with their relative contribution (b) over the year. The parameters of this simulation were set for a fish community of 35 mm with the seasonal variation of the environment (Fig.6).



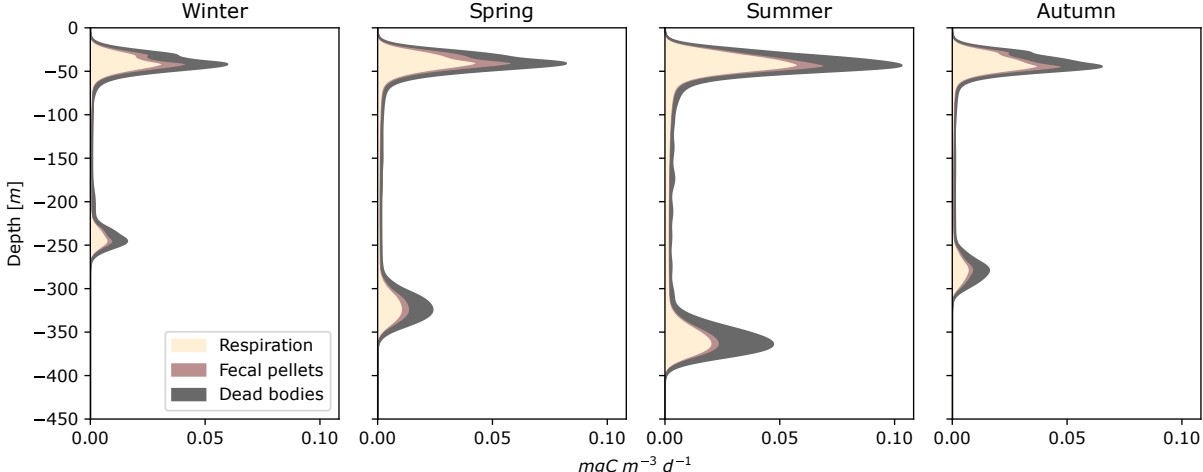

**Figure 8.** Seasonal variation of carbon detritus induced by micronekton at depth including the daily metabolic products ($D_m$), the daily production of fecal pellets ($D_g$) and dead bodies ($D_\mu$). This simulation was set for a fish community of 35 mm with the variation of the environment (Fig.6) as in Fig.7.

## 4 Discussion

In this work, we propose a bioenergetic model in order to explicit the importance of three carbon pathways including metabolic products, fecal pellets and dead bodies ($D_m$, $D_g$, $D_\mu$). The bioenergetic model implies a comprehensive consideration of all carbon pathways of micronekton including respiration rates that vary according to the type of activity. This allowed to represent the variation of the metabolic activities, i.e. resting, swimming and feeding, over time and across different depth layers (see Fig.S1 in the Supplement). Representation of DVMs allowed to estimate the amount of carbon that is produced continuously along the water column and their impact on micronekton's metabolism. Indeed, models that represent instantaneous migrations neglect the influence of this period on carbon transport. This amount of carbon produced during the DVM must be considered as it impacts the proportion of fecal pellets released at depth.

The formulation of the bioenergetic model showed that micronekton predominantly feed during crepuscular periods, exhibiting higher capture rates when they reach the surface at sunset and prior to their downward migration at dawn. This results in two daily peaks of activity due to the cost of swimming during DVM and to the cost of digestion, depending on the amount of ingested food at the surface (Fig.S1 in the Supplement). This is consistent with the results of Rosland and Giske (1997) where feeding rates are higher during crepuscular periods. During these periods, the light level is sufficient for locating prey while still low enough to avoid detection by predators. This supports the "anti-predation window" hypothesis, meaning that during dusk and dawn DVM minimizes the ratio of mortality risk to feeding rate (Clark and Levy, 1988). After being satiated at dawn, micronekton goes at depth avoiding visual predators during the day in the euphotic zone. Then, they rise to the top layers at dusk, getting hungry again as stated by the "hunger/satiation" hypothesis (Pearre, 2003).





### 4.1 Influence of micronekton's traits on carbon production

Considering the three main taxonomic groups of micronekton, significant differences in migration speed, daytime residence depth, and metabolic activity were observed among fish, crustaceans, and cephalopods (Fig.3). Fish emerged as the group exerting the greatest influence on the efficiency of carbon transport at depth within the micronekton community. Metabolic

activity also exhibited significant dependence on organism size, with smaller organisms demonstrating higher energy demands while migrating and releasing particles at shallower depths. The results showed a non-linear relationship between the size and the metabolic products (Fig.3j,k,l). This phenomenon could be attributed to larger organisms expending more energy on migration, given their increased visibility and tendency to migrate to greater depths. Consequently, our findings suggest that an intermediate size dependent on the taxonomic groups of individuals would yield the highest carbon transport efficiency

in the mesopelagic zone. However, it is important to interpret these results cautiously, as they rely on parameters such as the swimming speed coefficient ($a_{swim}$), which may vary depending on the community composition. Nevertheless, these parameters can be calibrated using acoustic detection of scattering layers and associated size distributions.

     Cephalopods have a particularly active metabolism, with important respiration rates, 1.5 to 1.7 times higher than those of fish species (Ikeda, 2016). These fast-growing pelagic organisms produce relatively low amounts of fecal pellets and have

short lifespan cycles (Boyle and Rodhouse, 2008). The respiration of DIC represented on average 60.5% of their production of carbon compared with 50% for fish and 39.2% for crustaceans (Fig.4, Table S1 in the Supplement). Likewise, the production of fecal pellets accounted on average for only 9.6% of carbon production compared with 21% for fish and 26% for crustaceans. Despite their smaller contribution to fecal pellet production and carbon excretion, cephalopods play a significant role in carbon cycling due to their elevated respiration rates. Understanding these metabolic differences among micronekton taxa is crucial

for accurately modeling carbon transport and ecosystem dynamics in the pelagic environment.

     Our findings indicate that a small subset of parameters significantly contributes to the variability in micronekton-induced carbon production (Fig.5). A recent study revealed that considering the range of plausible parameter values, carbon fluxes exhibited a six-fold variation (McMonagle et al., 2023). Specifically, the respiration coefficients emerged as the primary source of uncertainty in bioenergetic models for fish, consistent with prior research (Davison et al., 2013; McMonagle et al., 2023). Our

results corroborate this observation for crustaceans and cephalopods as well. To adequately account for micronekton's impact on the BCP, it is imperative to intensify field measurements of respiratory rates across various oceanic regions. The parameters exhibited a similar influence on both the biomass concentration of micronekton, production of POC and efficiency of carbon transport. Indeed, metabolic activity is directly dependent on biomass whereas it is the most sensitive inputs of models, and global estimates are highly uncertain. Others factors influenced the export efficiency of carbon. They are directly linked to

swimming speed, which depends mainly on factors including the $a_{swim}$ coefficient, as well as the assimilation efficiency, and evacuation rate for crustaceans (Fig.5). The digestion time of crustaceans is faster than the other groups and therefore has a greater effect on the proportion of fecal pellets released at depth.



## 4.2 Influence of environmental parameters on carbon production

Fish was the most effective taxonomic group for exporting carbon at depth. This group was therefore selected to carry out
analyses on the influence of environmental conditions on carbon export generated by migrating organisms.

Considering separately and simultaneously the seasonal variation of environmental parameters (Fig.S4 in the Supplement, Fig.7), the results showed that primary production drives the amount of carbon that will be exported and has a direct effect on light attenuation in the water column (Fig.S5 in the Supplement). Indeed, Irigoien et al. (2014) showed a strong relationship between primary production and fish biomass estimated from acoustic observations. During the bloom, productive surface waters increase the density of micronekton's preys but also limit the capture efficiency by reducing the water clarity (Fig.S5 in the Supplement). This resulted in moderate carbon concentrations induced by micronekton in spring, followed by a peak of carbon production in June (Fig.7). Thus, the link between light and PP is a key process in seasonal carbon export by migrant organisms.

Models of aquatic visual feeding showed that daylight variability is more important than prey abundance for predation efficiency (Aksnes and Giske, 1993; Aksnes and Palm, 1997). Irigoien et al. (2014) found that in clear oligotrophic waters, the energy transfer efficiency from phytoplankton to mesopelagic fish was higher than previously established. This may explain in our study's temperate region, a higher peak of micronekton concentrations in summer. Although the resource is less abundant than in spring, visual predation is more efficient thanks to clearer waters. Thus, micronekton may have a greater impact on the BCP during summer than in spring, when the carbon flux is mostly driven by the gravitational pump (e.g., Sanders et al., 2014). During summer, the resource's density diminished but their growth rate and the micronekton's capture rate are higher than in spring. Moreover, micronekton organisms migrate to deeper water where they released particles (Fig.8). However, the production of fecal pellets is not much higher than in spring. This is likely due to higher temperatures increasing the remineralization rate of organic particles.

In the model, temperature indirectly affects the primary production, estimated from satellite data, by influencing phytoplankton growth and nutrients availability. Higher temperatures can decrease the efficiency of the BCP by increasing the remineralization of organic carbon through respiration. Phytoplankton near their optimum temperature are very sensitive to warming, potentially disrupting their community composition and phenology (Middelburg, 2019). Warmer waters are expected to alter food-web structures by increasing plankton diversity in the North Atlantic and reducing copepod mean size (Beaugrand et al., 2010), which will subsequently affect micronekton biomass.

Our estimates of the seasonal variation of carbon detritus in the mesopelagic zone (Fig.8) are consistent with the findings of Bianchi et al. (2013) for different areas. In our study, winter carbon production is similar to estimates under oligotrophic conditions in the Pacific, around 0.6 mmol C m$^{-3}$ y$^{-1}$. Conversely, our summer carbon production estimates are close to those in productive regions, reaching 1.5 mmol C m$^{-3}$ y$^{-1}$. Their study suggested that carbon production in the mesopelagic zone by DVMs can contribute up to 40% of carbon export.



### 4.3 Model's limits

A model can provide a simplified representation of a complex system, allowing to gain insights into the fundamental mechanisms driving system behavior. By focusing on a reasonable amount of parameters, our model helps to elucidate important processes involved in the transport of carbon by micronekton. We discuss in this section the main choices made to simplify the representation of vertical dynamics, interactions between micronekton and their prey, and the energy budget of these organisms.

The dynamic of the DVM was based on the rate of change hypothesis, i.e. the migration speed of the organisms is dependent of the gradient of light (e.g., Andersen and Nival, 1991). However, other explanations have received great attention including the light preferendum hypothesis where organism follow an isolume (e.g., Cohen and Forward Jr, 2009) or adjust their position to remain in a "light comfort zone" (e.g., Frank and Widder, 2002; Røstad et al., 2016; Langbehn et al., 2019). Nevertheless, our method allows us to reproduce consistent migration speeds and daytime depth residence estimated from sound scattering

layers recorded at sea (e.g., Bianchi and Mislan, 2016; Cade and Benoit-Bird, 2015).

A key aspect of prey-predator relationships is how variations in prey quality can affect the amount of carbon assimilated per feeding period by individual micronekton. This approach offers a more realistic estimation compared to assuming uniform ingestion rates across all species (Woodstock et al., 2022). By considering differences in prey taxa, we can more accurately capture feeding behaviors and ecological interactions within the micronekton community, enabling the study of inter-annual varia-

tions. However, further research is necessary to minimize uncertainties associated with these feeding parameters. Micronekton taxa were modeled independently to avoid considering interactions between them, including competition for resources. According to the competitive exclusion principle, multiple species cannot coexist in the same environment while feeding on the same resource (Gauze and Teissier, 1935). This would necessitate making trophic compartments more complex, which was not the aim of this study. In our parsimonious modeling approach, relative concentrations of fish, crustaceans, and cephalopods

were defined to represent micronekton's community. Similar micronekton's biomass taxonomic ratio that we used in Table 1 was found based on the preliminary results of the APERO cruise (Fig.S7 in the Supplement), conducted in our study area. The size classes measured from organisms collected by a mid-water micronekton trawl also fall within the same size ranges (Fig.S7). In addition, the size used for the seasonal simulations correspond to average size found for fish (Fig.S7). However, those results must be taken with caution as they are highly dependent on the selectivity of the sampling gear (Kwong et al.,

2018).

### 4.4 Perspectives

There is an urgent need to address uncertainties associated with the Mesopelagic Migratory Pump (MMP) in light of changing ocean conditions, linked to global warming and overfishing. This entails a deeper understanding of micronekton ecology, metabolism and their role in carbon transport, considering the risk of their potential exploitation as a resource (Schadeberg

et al., 2023).

We suggest using this model as a tool to estimate the active carbon transport induced by micronekton, across different regions. Parameters could be calibrated using data collected from trawl samples, acoustic sounders, zooplankton nets and



sediment traps. Additionally, recording environmental factors such as light, primary production, and temperature at depth could help, in order to capture the variability in the relative contribution of micronekton to the BCP. Establishing recurrent seasonal sampling and deploying calibrated instruments across different regions would facilitate comparisons of environmental changes. For instance, it might help in understanding the factors that control migration depths and dynamics. Size is probably not the only morphological trait influencing nighttime depth occupation. Mesopelagic ecosystems being characterized by dim light, organisms emitting their own bioluminescence for communication/predation can, for example, be one trait of interest (de Busserolles et al., 2014). Moreover, modeling migration dynamics including state of satiation of organisms based on stomach content analyses sampled from trawl net sampling in layers identified by acoustics would improve the modeling of the vertical migration behavior. This implies that organisms would ascend only when their guts are empty. Indeed, not all organisms of the same species and size class rise to the surface at night (e.g., Bos et al., 2021).

Further studies on the BCP should investigate the role of gelatinous organisms, particularly salps, which are often overlooked in carbon budget assessments. Salps are known to filter large amounts of suspended particles and produce fast-sinking fecal pellets, potentially enhancing the efficiency of the BCP in the ocean (Phillips et al., 2009; Steinberg et al., 2023; Décima et al., 2023). During the APERO cruise, gelatinous represented on average 14% of the biomass collected by a mid-water trawls. However, we did not considered this group in this study, since they do not consume the same resource as micronekton.

To conclude, incorporating energetic and functional approaches into carbon budget models will improve our ability to predict and mitigate the impacts of environmental changes on marine ecosystems and global carbon cycling.



*Code availability.*

The source code (written in Python) supporting this article is available via Github at
https://github.com/helene-thib/model_dvm_carbon.

*Data availability.* Data from the APERO cruise including trawl data is available from MEMERY Laurent, TAMBURINI Christian, GUIDI
Lionel (2023) APERO 2023 cruise, RV Thalassa, https://doi.org/10.17600/18000666

*Author contributions.* All the authors contributed to the conception of the study. H.T carried out model simulations based on an early version
of the model developed by J.A-S with technical support from J-C.P. HT wrote the article with contribution from all the co-authors.

*Competing interests.* Authors have declared that none of the authors has any competing interests.

*Acknowledgements.* The authors would like to thank the captain and crew of N/O 'Thalassa' (Flotte Océanographique Française) for their
help during the APERO cruise (http://dx.doi.org/10.17600/18000666).
HT would like to thank the Grantham Foundation and WHOI's Ocean Twilight Zone project for funding and hosting the OTZ Symposium,
providing a valuable state of the art in the Biodiversity, Ecology, and the Biological Carbon Pump in the Ocean Twilight Zone and insightful
discussions.
We acknowledge the staff of the "Cluster de calcul intensif HPC" platform of the OSU Institut PYTHEAS (Aix–Marseille Université,
INSU-CNRS) for providing the computing facilities for the sensitivity analysis.
The authors greatly acknowledge the Imaging and Taxonomy platform of MIO (Aix-Marseille Université) and its personal for material
support in APERO micronekton samples determination and measurements.

*Financial support.* This manuscript contributes to the APERO project funded by the National Research Agency under the grant APERO
[grant number ANR ANR-21-CE01-0027] and by the French LEFE-Cyber program (CNRS, INSU).



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
