# Peer review of "Modeling the contribution of micronekton diel vertical migrations to carbon export in the mesopelagic zone"

_EGUsphere, 2024_

## Author Comment (AC1)

The authors would like to express their gratitude to the reviewer for the time and effort dedicated to evaluating the manuscript with insightful comments and suggestions.

This study analyzes the role of diurnal vertical migrations performed by micronekton (fish, crutaceans and cephalopods) on the biological carbon pump. The authors developed a simple model describing explicitly the vertical movements of the animals as well as ingestion, respiration and the production of fecal pellets. The model relies on three state variables which are the biomass of the preys, i.e. mesozooplankton which do not perform DVM, the biomass of the consumers and the gut content. The latter variable is necessary to accurately describe the production of fecal pellets. The consumers are visual predators meaning that they need light to capture their preys. During the night, they reside near the surface to feed. At dawn and dusk, they swim to stay at depth during the day so that they escape predation from their visual predators. In the model, temporal variation in the light levels triggers DVM. The model is run in constant as well as seasonnally varying environmental conditions. It is used to explore the role of size and taxonomy on the DVM patterns and its impacts on the active vertical transport of carbon. Detailed sensitivity analyses are performed by systematically exploring the parameters space. The main findings are: (1) in the temperate regions, DVM is responsible for an important vertical transport of carbon from the euphotic zone to the mesopelagic domain; (2) Size and taxonomy play a big role in driving this transport; (3) There are strong seasonal variations of this active export of carbon with a maximum reached in summer; (4) Results are very sensitive to some parameters, such as the vertical swimming speed, the metabolic rate and its sensitivity to temperature.

This study addresses an important topic which is increaslingly acknowledged as an important component of the biological carbon pump. The modeling framework is relevant as it includes the essential aspects of the animal physiology and it explicitely simulates the vertical movement of the organisms. It remains simple enough so that detailed sensitivity analyses remain feasible and can be applied to a large range of environmental conditions, except probably in the polar regions (polar days and nights). It is well written, even if as a non-english native speaker, I am not necessarily the best person to judge this aspect. Supplementary materials are interesting and bring some important additional information to the manuscript. However, I have some serious issues with the study, mainly with the model description and modeling assumptions. As the code is available on a github server, I have closely inspected it to check what is stated in the manuscript.

First, there is an issue in the system of equations 1. Specific dynamic action is a respiration term and thus, represents a loss of carbon. However, this is not included in the system despite the fact that it is routed to inorganic nutrients (equations 6). I checked the code to see if this corresponds to a bug but this is not the case as it is properly taken into account in the temporal evolution of the C biomass. This should be corrected since before I checked the code, I thought that the manuscript was relying on bugged results.

**Answer:**

We thank the reviewer for pointing out this mistake. There was a problem in writing the system of equations 1 as we did not included the specific dynamic action in the metabolic products. This term was well included in the code and therefore it did not affect the results. We replaced the term $mC$ by $Dm$ (Eq.6), which includes now all the metabolic products:

$$
\begin{cases}
\dfrac{\partial P}{\partial t} = \rho P\left(1 - \dfrac{P}{K(z)}\right) - \dfrac{\alpha_v(t,z)P}{1+\beta P}C \\[2ex]
\dfrac{\partial G}{\partial t} = -\dfrac{\partial(wG)}{\partial z} + \dfrac{\alpha_v(t,z)P}{1+\beta P}C - (d+\mu)G \\[2ex]
\dfrac{\partial C}{\partial t} = -\dfrac{\partial(wC)}{\partial z} + edG - D_m - \mu C
\end{cases}
$$

Second, the daily evolution of light is said to be described by equation 1 of the supplementary materials. This equation is difficult to read because the exponent n should apply to the sin function and not to wt. Furthermore, this equation implies that light levels are zero at night meaning that visual predators cannot feed at night when they stay close to the surface. The only available temporal window in that case is during dusk and dawn (about 2-3 hours a day). However, during that period, they move either up or down which implies that they are not colocated with their preys. Thus, according to that equation, they should not really be able to feed and they should starve and die. Yet, this is not the case. I was also disturbed by figure 2 which shows the daily evolution of light at the surface. On that figure, the relative light level during the night is not zero as it should be according to the equation displayed in the supplementary materials but somewhere between 0 and 0.6. I inspected the code and saw indeed that the actual coded equation is not exactly that of the supplementary materials but rather equation 1 to which a constant 0.5 is added. This explains why the consumers are able to survive in the model since with this 0.5, they are then able to feed at night. However, this 0.5 implies that light level during the night is about one third of its value during the day, which is obviously not correct! In addition, the Beer-Lambert equation is used with an attenuation coefficient of 0.001 m^(-1), meaning an attenuation depth of 1000 m! In very clear water, the coefficient is rather of the order 0.02-0.03, which is at least one order of magnitude higher ... This is not a typo because this 0.001 is the actual value used in the code. And the option to use a chlorophyll dependant coefficient is not activated and not included in the call to the beer-lambert function. To conclude with my issues on light, once in the manuscript and once in the supplementary materials, the authors say that day length at the winter solstice is 6 hours and is 12 hours at the summer solstice. This is obviously incorrect as day length at the summer solstice should be 18 hours.

**Answer:**

As mentioned by both referees, we agree that the method used to model surface irradiance needs to be clarified. In addition, we have made adjustments to allow micronekton to feed more efficiently during twilight periods, when it is not migrating.

To model daily irradiance, we propose to use a sinusoidal curve. For the gradient of surface irradiance (cf Eq.S1), which allows us to calculate migration speed, we now set a threshold: when micronekton reach the surface, they begin migrating once the light gradient falls below this threshold. The threshold has been set at 0.1 and the sensitivity of this parameter on carbon production will be investigated in the sensitivity analysis.

We also added a constant, that we now call $I_{min}$, to prevent the light level from reaching zero. As suggested by both referees, we will clarify this point in the supplementary material. For the computation of the capture rate, this constant must not be too high. We thus have set $I_{min}$

to 0.01, so that micronekton can still feed at night but much less efficiently than during the feeding windows.

The equation S1 is now written:

$$I_0(t) = 1 + I_{min} - \exp(-a \, sin^n(\omega t))$$

with $I_{min}$=0.01, $a$=4 and $n$ varies according to the day of the year.

Here is an example with the new formulation of the surface irradiance and the resulting migration pattern as in Figure 2:

[Figure]

We will also set a different coefficient of attenuation based on field data. We already implemented in the seasonal scenario in Figure 7 and 8 a coefficient of attenuation that varies according to the concentrations in chlorophyll a (Chl-a) (Eq.S5). According to this equation, the coefficient of attenuation varies between 0.04 and 0.07. We will use a value between this range for the simulations that do not include seasonal variations of the environment.

Third, in the supplementary materials, the authors shows results from some sensitivity results on the seasonal variations of temperature, light and PP (which is in fact phytoplankton biomass rather than PP). This is very interesting. Yet, I don't understant the changes they impose on the parameter C_alpha. In scenarios 1 and 2, this parameter is set to 3 but when light is seasonally varying (scenarios 3 and 4), it is set to values that are between 2 and 3 orders of magnitude lower. Yet, the resulting detritus biomass is similar. Furthermore, the chosen values in that second case are not consistant with figure S8. Weird!

**Answer:**

We agree with the referee that the term phytoplankton biomass is more appropriate than PP. It will be modified accordingly in the text.

The objectives of Figure S4 in the supplement material is not to present results corresponding to realistic scenarios. We aimed at testing the sensivity of seasonal variations of temperature,

light and phytoplankton biomass on carbon production. Therefore,, we used C_alpha as a scaling parameter to ensure a consistent prey/predator dynamic, i.e. to prevent consumers and resource concentrations from collapsing. This is why the carbon biomass is similar, as we were interested in the differences in annual variability between the scenarios. This will be clarified in the Material and methods section.

The different values of the attenuation coefficient explain the important differences in C_alpha values. With the redefined values for the attenuation coefficient, the range of C_alpha values is now reduced (between 0.7 and 7).

Fourth, I don't understand why the authors have added a **remineralization term** in the fecal pellets equation (equations 6). Are the detritus pools prognostic variables or simply diagnostic variables? I did not check the code for that specific aspect. What do the authors call the production of fecal pellets? Is it (1-e)dG or (1-e)dG -r(T)D_g? Why including this remineralization on fecal pellets but not on dead organisms? Why only remineralization and not vertical sinking which, for organisms of that size, is way more important at controlling the concentration than remineralization? This needs to be explained and discussed. On the topic of production, I don't understand what is displayed in Figure 7a. If this is the temporal evolution of detritus production, then the units are wrong. If this is the temporal evolution of the detritus pools, this is impossible since the animal dead bodies and the inorganic pool do not have a sink term. This is also the case in Figure S4.

**Answer :**

We agree with the referee that it is not appropriate to add the remineralization term for the fecal pellets only. As our study focuses on the variability of carbon production by micronekton, we should not have included particle remineralization. We will correct the manuscript accordingly in equation 6 as followed,

$$\frac{\partial D_g}{\partial t} = (1-e)dG$$

In figure 7a and S4 we showed the temporal evolution of detritus production, so this should be expressed in mgC m$^{-3}$ d$^{-1}$. We thank the referee to point out this error.

And as a biogeochemist, I was a little bit disturbed by the definition of the pe-ratio which traditionnally is the ratio of the export at some depth over PP whereas here, this a ratio between a flux and biomass

**Answer:**

As the state variables (P, G, C) unit is inmgC m$^{-3}$, we have defined the efficiency of carbon transport as the proportion of carbon biomass exported under a certain depth over the phytoplankton biomass (mgC m$^{-3}$). We therefore agree that it we should be called something other than pe-ratio as it is a different metric. We propose to call it Micronekton Carbon Export ratio (MCE-ratio).

I had some more minor points but considering the main issues I listed above, I think they are not really relevant at this stage. I did not make any comments on the results and discussion because of the concerns I had on the model formulation which according to me, raises

questions about the validity of the results. Regarding the results, one intriguing observation is the occurrence of small, localized peaks in the production of dead bodies between the epipelagic zone and the depth at which organisms reside during the day. What could explain these peaks?

We thank the referee to point out this problem. We checked carefully the code and found that this is due to a numerical artefact linked to the numerical scheme: the small peaks disappear when using a smaller time step. To counteract this numerical issue, a smaller time step of 0.2h allows us avoiding this numerical anomaly, as shown in the following figures:

[Figure]

dt=0.8h                    dt=0.2h

In conclusion, while this study shows considerable potential, it is not ready for publication in its current form. The authors need to revise certain aspects of the model formulation and provide clearer justifications and descriptions of their choices to ensure that the results are robust enough to support the subsequent analysis.

---

## Author Comment (AC2)

The authors would like to express their gratitude to the reviewer for the insightful comments and valuable suggestions.

General comments

One of the less-studied components of the biological carbon pump is the MMP, which contributes significantly to carbon flux. While most research has focused on zooplankton, the role of micronekton in the present-day carbon budget remains poorly quantified. This study addresses an important gap in our understanding and is highly relevant to marine ecology. What stands out in the modeling approach is the incorporation of physiological traits specific to different taxonomic groups, adding a useful dimension to the analysis. However, I believe the model remains somewhat theoretical due to the lack of data for validation.

Incorporating net sampling and acoustic data, if available, would be crucial for calibrating and validating the model outputs. The authors mention the presence of trawl and acoustic sampling data from the APERO cruise, conducted in the same region during June and July, but it is unclear whether these data were used for model validation. This would be a key step to enhance the reliability of the results.

**Answer:**

We thank the referee for pointing out this aspect. Some of the data collected during the APERO cruise are useful for the calibration and validation of the model.

Regarding the calibration of the model, we have used the trawling data collected during the APERO cruise to calibrate the size classes and the taxonomic ratios, i.e. the relative biomass of fishes, cephalopods and crustaceans (Figure S7).

For the validation of our results, identifying species from acoustic data remains a challenge and this goes far beyond the scope of our study. Nevertheless, APERO acoustic and trawl data enabled us to characterize several migrating layers, moving from depths of 150-700 m during the day to approximately 40 m at night. These migrating layers sampled by trawling are mainly composed of a mix of fishes, cephalopods, euphausiids and decapods. We can therefore compare the simulated depths reached by micronekton taxa in our model with the depths observed in acoustic echograms. For example, migrant fish are mainly composed of myctophids. Similar to the findings of Watanabe et al., (1999), we found a maximum concentration of fish at 0-50 m during the night and at 400-500 m/600-800 m during the day, depending on the station. Here is one example of an echogram of a station of this APERO cruise showing several migration depths, with white lines representing the trawl's trajectory:

[Figure]

In addition, several myctophid species was reported in the literature with a depth stratification by size during the day (e.g. Frost and McCrone 1979). Quetglas et al. 2010 observed the same pattern for migrating species of the genus *Histioteuthis*, that were the main migrating cephalopods collected in trawls of the APERO cruise.

We will clarify this point and improve the manuscript accordingly.

I would also like to have clarifications for the choice of 200 m depth as the euphotic zone for calculating the efficiency of particulate organic carbon (POC) transport. Most studies typically use a depth of 100 m for the euphotic zone, so it's important to explain the reasoning behind your choice of depth. Was this depth based on specific data from the PAP-SO station, or was it taken from existing literature? The selection of the depth threshold for the euphotic zone is critical, as it can significantly affect the calculated efficiency of POC transport. If not well justified, using 200 m instead of 100 m could potentially skew the results, leading to over- or underestimation of the POC transport efficiency. The export efficiency is usually calculated using the flux at a specific depth over the net primary production, what does integration phytoplankton concentration in the surface layers represent? is it a biomass?

**Answer:**

We agree with the referee that this choice needs clarification. We chose 200m depth for the export of carbon as the average of the mixed layer depth (MLD) at PAP-SO. In the manuscript, we will therefore use MLD depth rather than euphotic zone. In addition, in Figure 7b, we will choose a more realistic approach as the MLD at this station varies annually between approximately 30 and 300 m (Hartman et al., 2012). We will then calculate the variability of the efficiency of carbon transport by including the annual variation in MLD as the export depth. However, this will not significantly change the dynamic of carbon export as the MLD is deep in winter, micronekton are found in shallower depth, and production is low.

As the state variables (P, G, C) unit is in mgC.m$^{-3}$, we have defined the efficiency of carbon transport as the proportion of carbon biomass exported under a certain depth over the phytoplankton biomass (mgC.m$^{-3}$).

We therefore agree that we it should be called something other than pe-ratio as it is a different metric. We propose to call it Micronekton Carbon Export ratio (MCE-ratio).

Although the current model is a simplified 1D water column setup, the authors made several assumptions and choices in their study's design to assess the role of the DVM of micronekton on the organic carbon budget. One significant assumption is that mesozooplankton do not migrate and are restricted to the epipelagic layer, which could influence the estimated carbon flux. Numerous studies, including those by Kiko et al. (2017, 2020), and Bianchi et al. (2013), have demonstrated that zooplankton also exhibit DVM and that they are usually present between 300–600 m. Incorporating a portion of zooplankton in the deeper layers of the model would be important, as they could serve as prey for micronekton, potentially contributing to fecal pellet production and, in turn, to carbon transport.

**Answer:**

We agree with the referee that this assumption is simplistic and needs to be justified. Stomach content analyses of selected micronekton organisms were performed during the APERO cruise. Migrating micronekton had undigested prey items only at the surface and not at depth during the night, where we found only digested food suggesting that prey items were ingested earlier in the day. Sameoto (1989) had similar findings for *Benthosema glaciale*, the most common mesopelagic fish specie found during the APERO cruise. We believe that they migrate to greater depths than zooplankton, notably because of their greater risk of being spotted by predators and their greater swimming ability. Therefore we did not model zooplankton diel vertical migration as this may not impact micronekton consumption.

Another issue concerns the environmental variables influencing DVM. For instance, oxygen concentration plays a crucial role, especially when micronekton inhabit oxygen minimum zones (OMZs). In such zones, low oxygen availability limits respiration and metabolic rates, impacting vertical migration behavior. However, the model assumes that micronekton feed exclusively at the surface, without considering the potential effects of hypoxic conditions on DVM patterns and metabolic processes, this might be due to the modeling of the PAP-SO station. But factoring in these environmental constraints could offer a more nuanced and accurate representation of micronekton's contribution to the carbon cycle and would make the model fit to be globally applied to a large range of environmental conditions such as the Atlantic OMZ.

**Answer:**

We agree that it would be necessary to add oxygen as a factor limiting migration depths under a certain threshold of hypoxia, in order to use this model at global scale. We will mention this aspect in the discussion.

We have selected the environmental variables that can significantly influence micronekton dynamics at PAP-SO station, where there is no such hypoxic conditions.

Another concern I have is with the light in the model. The equation in the supplementary materials assumes that surface irradiance is zero at night, which is not entirely accurate. Even at night, there is still some ambient light (e.g., from the moon), and predators that rely on visual predation may still be able to feed, albeit less effectively. The current model restricts feeding to dusk and dawn, which is problematic since micronekton are supposed to be migrating between surface and deeper layers during these periods. According to this assumption, predators would be unable to feed and, over time, would likely starve. However, when examining Figure 2 in the manuscript, it appears that surface light never actually

reaches zero, contradicting the assumption in the equation. This inconsistency suggests that something might be missing or oversimplified in the supplementary equation. Could it be that the equation does not fully account for low-light conditions at night? is there a missing parameter to this equation? Additionally, have you considered varying light levels to simulate different migration depths?

**Answer:**

As mentioned by both referees, we agree that the method used to model surface irradiance needs to be clarified. In addition, we made make adjustments to allow micronekton to feed more efficiently during twilight periods when it is not migrating.

To model daily irradiance, we propose to use a sinusoidal curve. For the gradient of surface irradiance (cf Eq.S1), which allows us to calculate migration speed, we now set a threshold: when micronekton reach the surface, they begin migrating once the light gradient falls below this threshold. The threshold has been set at 0.1 and the sensitivity of this parameter on carbon production will be investigated in the sensitivity analysis.

We also added a constant, that we now call $I_{min}$, to prevent the light level from reaching zero. As suggested by both referees, we will clarify this point in the supplementary material. For the computation of the capture rate, this constant must not be too high. We thus have set $I_{min}$ to 0.01, so that micronekton can still feed at night but much less efficiently than during the feeding windows.

A few small comments concern these two points: I am also unclear about the changes imposed on the parameter $C\alpha$ and why even though it varies between the scenarios we still have the same resulting detritus concentration. Could you provide further clarification on how the $c\alpha$ was modified and why such large differences do not lead to corresponding changes in detritus biomass? In equation 6, it is unclear to me, if remineralization only applies to fecal pellets or if the author considered it also in the dead organisms, considering that both processes would contribute to organic matter degradation, so excluding remineralization for dead bodies seems inconsistent.

**Answer:**

We thank the referees to point out this problem. As indicated in the reply to reviewer 1, the objectives of Figure S4 in the supplement material are not to present results corresponding to realistic scenarios. We aimed at testing the sensitivity of seasonal variations of temperature, light and phytoplankton biomass on carbon production. Therefore, we used C_alpha as a scaling parameter to ensure a consistent prey/predator dynamic, i.e. to prevent consumers and resource concentrations from collapsing. This is why the carbon biomass is similar, as we were interested in the differences in annual variability between the scenarios. This will be clarified in the Material and methods section.

The different values of the attenuation coefficient explain the important differences in C_alpha values. With the redefined values for the attenuation coefficient, the range of C_alpha values is now reduced (between 0.7 and 7).

We also agree that it is not pertinent to add this remineralization term only for the fecal pellets. As our study focuses on the variability of carbon production by micronekton, we

should not have included particle remineralization. We will correct the manuscript accordingly in equation 6 as follow,

$$\frac{\partial D_g}{\partial t} = (1-e)dG$$

We re-run the simulations for the seasonal scenario in Figure 7, taking into account the adjustments made to the model as presented in both answer to the referees. This includes the modification of surface irradiance, the threshold for migration speed, the non-remineralization of fecal pellets and the export depth varying annually with the MLD.

Here is the comparison between the old results on the left and the new ones on the right for the annual variation of the carbon production integrated along depth:

[Figure]

The carbon production and its dynamic remain similar, but the peak is slightly shifted.

The proportion of fecal pellets is also higher than the previous results as we removed the remineralization term.

In conclusion, while this study holds significant potential, it still needs additional work for it to be published. The article could benefit from improvements in the clarity of idea presentation and the explanations behind methodological choices. The authors should validate the model, as this is essential for ensuring the robustness of the results. They also need to refine the treatment of light, zooplankton migration, remineralization, and detritus dynamics, and resolve inconsistencies in parameter adjustments.

References: -Bianchi et al. (2013) doi:10.1002/gbc.20031

-Kiko et al. (2020), doi: 10.3389/fmars.2020.00358

-Kiko et al. (2017), DOI: 10.1038/NGEO3042

Frost, B. W., & McCrone, L. E. (1979). Vertical distribution, diel vertical migration, and abundance of some mesopelagic fishes in the eastern subarctic Pacific Ocean in summer. *Fish. Bull*, *76*(4), 751-770.

Hartman, S.E., Lampitt, R.S., Larkin, K.E., Pagnani, M., Campbell, J., Gkritzalis, T., Jiang, Z.-P., Pebody, C.A., Ruhl, H.A., Gooday, A.J., Bett, B.J., Billett, D.S.M., Provost, P., McLachlan, R., Turton, J.D., Lankester, S., 2012. The Porcupine Abyssal Plain fixed-point sustained observatory (PAP-SO): variations and trends from the Northeast Atlantic fixed-point time-series. ICES J. Mar. Sci. 69, 776–783.

Sameoto, D., 1989. Feeding ecology of the lantern fish Benthosema glaciale in a subarctic region. Polar Biol. 9, 169–178.

Watanabe, H., Kawaguchi, K., Ohno, A., 1999. Diel vertical migration of myctophid fishes (Family Myctophidae) in the transitional waters of the western North Pacific. Fish Ocean.

---

## Author Response (AR2)

Dear editor,

Please find attached all the documents requested for the revisions of the manuscript. This includes a revised version of the manuscript ("egusphere-2024-2074-manuscript-version3.pdf") and the supplement ("egusphere-2024-2074-supplement-version2.pdf"), a point-by-point reply to the comments ("egusphere-2024-2074-author_response-version1.pdf"), and a marked-up manuscript version showing the changes ("egusphere-2024-2074-ATC1.pdf").

This file provides a detailed, point-by-point response to the comments from the two referees, as a complement to the previous answers we provided during the open discussion, which can be found here:
Reply on referee 1 (RC1): https://doi.org/10.5194/egusphere-2024-2074-AC1
Reply on referee 2 (RC2): https://doi.org/10.5194/egusphere-2024-2074-AC2

The main revisions in the manuscript are shown here in green, while the explanations are provided in black.
It is accompanied by a marked-up version of the manuscript which highlights the changes. Text highlighted in blue indicates insertions or modifications, while text in green denotes sections that have been moved.

**1) Issue in the system of equations 1**

**RC1**: "Specific dynamic action is a respiration term and thus, represents a loss of carbon. However, this is not included in the system despite the fact that it is routed to inorganic nutrients."

There was a problem in writing the system of equations 1 as we did not include the specific dynamic action in the metabolic products of the consumers (C). We replaced the term *mC* by *Dm* (Eq.6), which includes all the metabolic products.

**- Line 109 Eq.1**:

$$\frac{\partial C}{\partial t} = -\frac{\partial(wC)}{\partial z} + edG - D_m - \mu C$$

**2) Issues in daily evolution of light in equation S1 of the supplement**

**RC1**: "This equation is difficult to read because the exponent n should apply to the sin function and not to wt. Furthermore, this equation implies that light levels are zero at night meaning that visual predators cannot feed at night when they stay close to the surface. The only available temporal window in that case is during dusk and dawn (about 2-3 hours a day). However, during that period, they move either up or down which implies that they are not colocated with their preys. Thus, according to that equation, they should not really be able to feed and they should starve and die. Yet, this is not the case. I was also disturbed by figure 2 which shows the daily evolution of light at the surface. On that figure, the relative light level during the night is not zero as it should be according to the equation displayed in the supplementary materials but somewhere between 0 and 0.6. I inspected the code and saw indeed that the actual coded equation is not exactly that of the supplementary materials but rather equation 1 to which a constant 0.5 is added. This explains why the consumers are able to survive in the model since with this 0.5, they are then able to feed at night. However, this 0.5 implies that light level during the night is about one third of its value during the day, which is obviously not correct! In addition, the Beer-Lambert equation is used with an attenuation coefficient of 0.001 m^(-1), meaning an attenuation depth of 1000 m! In very clear water, the coefficient is rather of the order 0.02-0.03, which is at least one order of magnitude higher ... This is not a typo because this 0.001 is the actual value used in the code. And the option to use a chlorophyll dependant coefficient is not activated and not included in the call to the beer-lambert function. To conclude with my issues on light, once in the manuscript and once in the

supplementary materials, the authors say that day length at the winter solstice is 6 hours and is 12 hours at the summer."

**RC2**: "The equation in the supplementary materials assumes that surface irradiance is zero at night, which is not entirely accurate. Even at night, there is still some ambient light (e.g., from the moon), and predators that rely on visual predation may still be able to feed, albeit less effectively. The current model restricts feeding to dusk and dawn, which is problematic since micronekton are supposed to be migrating between surface and deeper layers during these periods. According to this assumption, predators would be unable to feed and, over time, would likely starve. However, when examining Figure 2 in the manuscript, it appears that surface light never actually reaches zero, contradicting the assumption in the equation. This inconsistency suggests that something might be missing or oversimplified in the supplementary equation. Could it be that the equation does not fully account for low-light conditions at night? is there a missing parameter to this equation? Additionally, have you considered varying light levels to simulate different migration depths?"

We implemented in the section *Vertical distribution modeling* of the *Material and methods* and in the section *Modeling irradiance* of the Supplement, the modifications of the formulation of the daily irradiance ($I_0$), the swimming speed (w) and the visual capture rate ($\alpha_v$).

To model daily irradiance, we used a sinusoidal curve. For the gradient of surface irradiance (cf Eq.S1), which allows us to calculate migration speed, we now set a threshold ($\Delta_{mig}$): when micronekton reaches the surface, they begin migrating once the light gradient falls below this threshold. The threshold is set to 0.1 and the sensitivity of this parameter on carbon production is investigated in the sensitivity analysis (Fig.5).

We also added a constant $I_{min}$, to prevent the light level from reaching zero (Eq.S1).

**- Section *Modeling irradiance* of the Supplement:**

Surface irradiance ($I_0$ in Eq.2,4) was modeled as a periodic function of time $t$, varying over the day as follows,

$$I_0(t) = (I_{min} + I_{max}) - I_{max} \exp(-a \, sin^n(\omega t)) \tag{1}$$

with $I_{min} = 0.01$ and $I_{max} = 1$, the minimum and maximum level of light, $\omega = (2\pi)/2H$ where $H$=24h, the parameter $n$=15, defining the timing of twilight hours, and the parameter $a = 4$ defining the degree of flattening of the curve (see an example in Fig.2).

[Figure]

Figure S4: Relative daily migration speed for a fish measuring 35mm. A positive swimming speed causes organisms to go down to the bottom of the water column, and a negative speed causes them to rise to the surface.

**- Lines 125-132:**

The swimming speed ($w$ in m h$^{-1}$) is assumed to depend on the swimming abilities of the migrant organisms, their size and the gradient of surface irradiance ($I_0$, modeled in the Supplement). Migrant organisms leave the surface only when the light gradient exceeds a threshold ($\Delta_{mig}$=0.1), allowing them to feed more efficiently at sunrise, before descending to depth and before sunset when they return to the surface. According to these assumptions, the swimming speed during the day is modeled as follows,

$$w(t) = \begin{cases} \frac{w_0}{I_0(t)} \frac{dI_0}{dt}, & \text{if } dI_0 > \Delta_{mig} \text{ and } z = z_{min} \\ 0, & \text{otherwise} \end{cases}$$

where $w_0$ represents the maximum swimming speed during the migration phases ($w_{max}$) normalized by the maximum light gradient, and $z_{min}$ is the minimum depth at which the maximum abundance of C occurs. The maximum swimming speed is function of $a_{swim}$, the swimming coefficient depending on the taxonomic group (see Table.2) and $L$ the body length (cm),

$$w_{max} = a_{swim}\ L \tag{2}$$

**- Line 137 and Eq.S5**: We defined a different coefficient of attenuation based on field data ($\psi$=0.05). We already implemented in the seasonal scenario in Figure 7 and 8 a coefficient of attenuation that varies according to the concentrations in chlorophyll a (Chl-a) (Eq.S5). According to this equation, the coefficient of attenuation varies between 0.04 and 0.07. We changed the value of 0.001 by a value between this range for the simulations that do not include seasonal variations of the environment.

- **Line 434**: We specified that the variation in migration depth shown in Figure 8 is due to seasonal changes in light levels.

**3) Calibration of the coefficient $c_\alpha$**

**RC1:** "I don't understand the changes they impose on the parameter C_alpha. In scenarios 1 and 2, this parameter is set to 3 but when light is seasonally varying (scenarios 3 and 4), it is set to values that are between 2 and 3 orders of magnitude lower. Yet, the resulting detritus biomass is similar. Furthermore, the chosen values in that second case are not consistent with figure S8."

**RC2:** "I am also unclear about the changes imposed on the parameter $C\alpha$ and why even though it varies between the scenarios we still have the same resulting detritus concentration."

The calibration of the coefficient $c_\alpha$ for the different simulations, involving different average community size, taxonomy (fish, crustacean or cephalopod) and environmental conditions, needed clarification as mentioned by both referees.

The calibration of this coefficient is explained **lines 269-275**. As we changed some aspects of the model including the coefficient of attenuation ($\psi$) affecting the capture rate ($\alpha_v$), we redefined new values of $c_\alpha$ that are listed in the Supplement in **Table S3** for the seasonal simulations and in **Table S4** for the simulations involving different size and taxonomic group.

The capture rate coefficient ($c_\alpha$) is calibrated to ensure a consistent transfer efficiency ($\gamma$) of 10% between the biomass of consumers and their resource (Table.1). This transfer efficiency is further adjusted using a taxonomic ratio ($\delta$), which allows for the estimation of the relative biomass distribution among fishes, crustaceans, and cephalopods within the micronekton community.

Therefore, the value of $c_\alpha$ is estimated for each simulation, allowing the calibration for P and C concentrations. This calibration varies according to size, taxonomic group, and seasonal simulation. The values used in each simulations are provided in the Supplement (see Table S3,S4).

**4) Remineralization term in the fecal pellets equation**

**RC1**: "I don't understand why the authors have added a **remineralization term** in the fecal pellets equation (equations 6). Are the detritus pools prognostic variables or simply diagnostic variables?"

**RC2**: "In equation 6, it is unclear to me, if remineralization only applies to fecal pellets or if the author considered it also in the dead organisms, considering that both processes would contribute to organic matter degradation, so excluding remineralization for dead bodies seems inconsistent."

**Line 145 Eq.5**: We corrected the equation of fecal pellets production as followed to not include the remineralization term, as discussed by both referees.

$$\frac{\partial D_g}{\partial t} = (1 - e)dG$$

**5) Wrong unit of production**

**RC1**: "On the topic of production, I don't understand what is displayed in Figure 7a. If this is the temporal evolution of detritus production, then the units are wrong. If this is the temporal evolution of the detritus pools, this is impossible since the animal dead bodies and the inorganic pool do not have a sink term. This is also the case in Figure S4."

**Fig.7a and Fig.S5**: As mentioned by RC1, in figure 7a and S5 we show the temporal evolution of the daily detritus production. We corrected the unit in mgC m$^{-2}$ d$^{-1}$.

**6) Definition of the pe-ratio**

**RC1:** "I was a little bit disturbed by the definition of the pe-ratio which traditionally is the ratio of the export at some depth over PP whereas here, this a ratio between a flux and biomass."

**RC2**: "I would also like to have clarifications for the choice of 200 m depth as the euphotic zone for calculating the efficiency of particulate organic carbon (POC) transport. Most studies typically use a depth of 100 m for the euphotic zone, so it's important to explain the reasoning behind your choice of depth. Was this depth based on specific data from the PAP-SO station, or was it taken from existing literature?

The export efficiency is usually calculated using the flux at a specific depth over the net primary production, what does integration phytoplankton concentration in the surface layers represent? is it a biomass?"

We defined the efficiency of carbon transport as the proportion of carbon biomass exported below a specified depth, referred to as the export depth, situated beneath the mixed layer depth (MLD), relative to the phytoplankton biomass (mgC m$^{-3}$). Initially termed the pe-ratio, this metric was renamed as it represents a distinct measure. We now refer to it as the Micronekton Carbon Export ratio (MCE-ratio). In **Fig.3 and Fig.5**, the MCE-ratio is calculated below 200 m, using an annual average MLD value. In **Fig.7**, we incorporate the seasonal variation of the MLD at PAP-SO to compute the MCE-ratio.

- **Lines 222-227:**

The mixed layer depth (MLD) is generated using monthly mean data from the Atlantic-Iberian Biscay Irish- Ocean Physics Analysis and Forecast (E.U. Copernicus Marine Service Information. Marine Data Store. https://doi.org/10.48670/moi-00027). The MLD was considered as the export depth to calculate the efficiency of POC transport by micronekton for simulations considering seasonal variations of the environment. The Micronekton Carbon Export ratio ($MCE-ratio$) was then computed as the integrated biomass of fecal pellets and dead bodies under the MLD divided by the integrated phytoplankton biomass in the surface layers.

- **Lines 235-237:**

prising fecal pellets and dead bodies that sediment and sequester carbon at depth in the water column, and 3) the efficiency of POC transport below 200 m, as the average annual MLD at PAP-SO ($MCE-ratio_{200}$). This metric is calculated as the total production of POC under 200 m depth divided by the integrated phytoplankton biomass in the surface layers. This allows

**7) Small peaks in the production of dead bodies**

**RC1**: "Regarding the results, one intriguing observation is the occurrence of small, localized peaks in the production of dead bodies between the epipelagic zone and the depth at which organisms reside during the day. What could explain these peaks?"

**Line 124 and Fig.3**: We used a smaller time step (dt=0.1) than the previous one (dt=0.8), to avoid the intermediate small peaks that were observed in Fig.3, regarding the production of dead bodies.

**8) Validation and calibration of the model with trawl and acoustic data**

**RC2**: " Incorporating net sampling and acoustic data, if available, would be crucial for calibrating and validating the model outputs. The authors mention the presence of trawl and acoustic sampling data from the APERO cruise, conducted in the same region during June and July, but it is unclear whether these data were used for model validation."

**Lines 380-387**: We discussed about the validation of the model for the depth ranges showed in Fig.3.

Fig.3 illustrates that migrant organisms exhibit depth stratification with size, as swimming speed is proportional to body length (see Eq.2). This pattern has been observed across various taxa, that were collected by a mid-water trawl deployed during the APERO cruise, targeting scattering layers detected by an echosounder (Fig.S9). Migrant layers displayed peak abundances at 0-50m at night and 400-800m during the day, depending on the station, consistent with depth ranges found by Watanabe et al. (1999). Depth stratification by size was observed in several myctophid species (e.g., Badcock and Merrett, 1976), sergestid shrimps (e.g., Flock and Hopkins, 1992; Koukouras et al., 2000; Vestheim and Kaartvedt, 2009) and cephalopods such as *Histioteuthis* squids (Quetglas et al., 2010). Indeed, larger animals descend deeper to avoid predation and thrive in colder, nutrient-poor waters due to their lower mass-specific metabolic rates.

**9) DVM of zooplankton**

**RC2:** "One significant assumption is that mesozooplankton do not migrate and are restricted to the epipelagic layer, which could influence the estimated carbon flux. Numerous studies, including those by Kiko et al. (2017, 2020), and Bianchi et al. (2013), have demonstrated that zooplankton also exhibit DVM and that they are usually present between 300–600 m. Incorporating a portion of zooplankton in the deeper layers of the model would be important, as they could serve as prey for micronekton, potentially contributing to fecal pellet production and, in turn, to carbon transport."

**Lines 475-480**: As pointed out by the second referee, we clarified why we did not include the DVM of zooplankton.

Another factor influencing micronekton feeding behavior is the spatial distribution of their prey. Analysis of gut contents from migratory species during the APERO cruise suggested that these organisms primarily feed at the surface (data unshown), leading us to constrain their resources to this layer. While zooplankton is known to exhibit

DVM (e.g., Bianchi et al., 2013; Kiko et al., 2020), they typically migrate to relatively shallow depths. As a result, we did not incorporate zooplankton DVM into the model, as it is unlikely to significantly affect micronekton consumption patterns.

**10) Oxygen limitation**

**RC2**: "For instance, oxygen concentration plays a crucial role, especially when micronekton inhabit oxygen minimum zones (OMZs). In such zones, low oxygen availability limits respiration and metabolic rates, impacting vertical migration behavior. However, the model assumes that micronekton feed exclusively at the surface, without considering the potential effects of hypoxic conditions on DVM patterns and metabolic processes, this might be due to the modeling of the PAP-SO station."

**Lines 455-461**: We discussed about the importance of considering oxygen limitation when modeling DVM, in order to use the model at a global scale, as pointed out by the second referee.

Light is a key factor in avoiding visual predators, but oxygen gradients may play a stronger role, particularly in regions with oxygen minimum zone (OMZ) (Bianchi et al., 2013). In regions without OMZs, migration depths are primarily controlled by light levels, but to achieve a global perspective, our model should incorporate co-limitation by both oxygen and light. Nevertheless, our method allowed us to reproduce consistent migration speeds and daytime depth residence in a non-hypoxic region, as inferred from sound scattering layers data recorded at sea, with an approximate migration duration of 2 hours (e.g., Bianchi and Mislan, 2016; Cade and Benoit-Bird, 2015).